



# Long-term air concentrations, wet deposition, and scavenging ratios of inorganic ions, HNO₃ and SO₂ and assessment of aerosol and precipitation acidity at Canadian rural locations

Irene Cheng and Leiming Zhang

Air Quality Research Division, Science and Technology Branch, Environment and Climate Change Canada, 4905 Dufferin Street, Toronto, Ontario, M3H 5T4, Canada

*Correspondence to*: Irene Cheng (irene.cheng@canada.ca)

**Abstract.** This study analyzed long-term air concentrations and annual wet deposition of inorganic ions and aerosol and precipitation acidity at 30 Canadian sites from 1983-2011. Scavenging ratios of inorganic ions and relative contributions of

particulate- and gas-phase species to $NH_4^+$, $NO_3^-$, and $SO_4^{2-}$ wet deposition were determined. Long-term median atmospheric $NH_4^+$, $NO_3^-$, and $SO_4^{2-}$ between sites ranged from 0.1-1.7, 0.03-2.0, and 0.6-3.5 μg m$^{-3}$, respectively. Their median annual wet deposition varied from 0.2-5.8, 0.8-23.3, and 0.8-26.6 kg ha$^{-1}$ a$^{-1}$. Geographical patterns of atmospheric $Ca^{2+}$, $Na^+$, $Cl^-$, $NH_4^+$, $NO_3^-$, and $SO_4^{2-}$ were similar to wet deposition and attributed to anthropogenic sources, sea-salt emissions, and agricultural emissions. Decreasing trends in atmospheric $NH_4^+$ (1994-2010) and $SO_4^{2-}$ (1983-2010) were prevalent.

Atmospheric $NO_3^-$ increased from 1991-2001 and declined from 2001-2010. These results are consistent with SO₂, NOₓ and NH₃ emission trends in Canada and the U.S. Widespread declines in annual $NO_3^-$ and $SO_4^{2-}$ wet deposition ranged from 0.07-1.0 kg ha$^{-1}$ a$^{-1}$ (1984-2011). Acidic aerosols and precipitation impacted southern and eastern Canada more than western Canada; however both trends have been decreasing since 1994. Scavenging ratios of particulate $NH_4^+$, $SO_4^{2-}$ and $NO_3^-$ differed from literature values by 22%, 44% and a factor of 6, respectively, because of the exclusion of gas scavenging.

Average gas and particle scavenging contributions to wet $NO_3^-$ deposition were 72±23% for HNO₃ and 28±23% for particulate $NO_3^-$. SO₂ and particulate $SO_4^{2-}$ contributed 37±20% and 63±20% to wet $SO_4^{2-}$ deposition, respectively. NH₃ and particulate $NH_4^+$ contributed 30±19% and 70±19% to wet $NH_4^+$ deposition.



# 1 Introduction

The Canadian Air and Precipitation Monitoring Network (CAPMoN) measures trace gas concentrations and particulate inorganic ion concentrations in air and precipitation at rural locations across Canada. Since 1983, the network has been collecting filter and precipitation samples and the number of sites has expanded to 33 as of 2010. CAPMoN was developed to monitor trends in atmospheric pollutants contributing to smog and acid rain, and the data was later used to assess the impacts of environmental policies in the Canada-U.S. Air Quality Agreement. This bilateral agreement signed in 1991 recognizes the impacts of transboundary pollution and sets objectives to reduce $SO_2$ and $NO_x$ emissions.

In this study, the focus is on the particulate base cations ($Ca^{2+}$, $Mg^{2+}$, $K^+$, $Na^+$, $NH_4^+$) and acidic anions ($Cl^-$, $NO_3^-$, and $SO_4^{2-}$), nitric acid, and sulfur dioxide that have direct impacts on acid rain. Nitrates and sulfates in acid rain reduce soil quality by causing the depletion of base cations, which are plant nutrients and are also involved in neutralizing acids. Base cations in soil can be replenished by mineral weathering, deposition, wind erosion, agricultural tilling, and forest fires (Hedin et al., 1994; Driscoll et al., 2001). However, when acidic deposition exceeds the supply of base cations, soil acidification occurs. Soil acidity has consequently increased the leaching of inorganic aluminum (Al) monomers, which is a toxic form of Al to plants and animals (Driscoll et al., 2001). Trees (e.g., red spruces and sugar maples) experienced damage to foliage, decreased adaptability to cold climates, slower growth, and mortality during 1960s-1980s from direct and indirect impacts of acid rain (Driscoll et al., 2001; Watmough and Dillon, 2003). Acid rain and runoff of acidic soil also increased nitrates, sulfates and inorganic Al and reduced pH in surface waters of Atlantic Canada, southcentral Ontario, and northeastern U.S. (Clair et al., 2002; Driscoll et al., 2003; Jeffries et al., 2003). Lake acidification has led to detrimental effects including mortality on zooplankton and fish (Driscoll et al., 2001 and references therein). Terrestrial birds are also impacted because when calcium is depleted from soil, less calcium-rich insects are available for birds to consume (Hames et al., 2002). Calcium deficiency in birds can cause eggshell thinning and other reproductive consequences (Hames et al., 2002).

Assessments of lake acidification in the previous decade indicate declines in nitrates and sulfates in surface water, some improvements to pH and acid neutralizing capacity, and conversion to less toxic organic Al (Clair et al., 2002; Driscoll et al., 2003; Jeffries et al., 2003; Kothawala et al., 2011; Strock et al., 2014). Although nitrate and sulfate deposition have been decreasing, surface water conditions have not improved at the same rate because nitrates and sulfates that have accumulated in soil and wetlands over a long period of time is gradually releasing to surface waters (Stoddard et al., 1999; Driscoll et al., 2001; Clair et al., 2002; Jeffries et al., 2003). A recent assessment by Lawrence et al. (2015) indicates no additional soil acidification and that acid deposition effects on soil have started to diminish in northeastern U.S. and eastern Canada according to indicators, such as exchangeable Ca and Al, base cations, and pH levels. Considering the role of inorganic ions on acid deposition effects on biota, it is important to continually study the wet deposition of inorganic ions.



The wet deposition of particulate base cations and acidic anions depend on the particulate concentrations of these inorganic ions in air and some trace gases, such as nitric acid and sulfur dioxide. This simplified relationship is the premise behind the scavenging ratio, defined as a ratio of a pollutant's concentration in precipitation to that in air. In reality, wet deposition is a very complex process that involves an understanding of cloud and precipitation processes and aqueous phase chemistry, which are considered to be the major sources of uncertainty in wet deposition modeling (Tost et al., 2007; Kajino and Aikawa, 2015). Scavenging ratios can be considered a measure of the wet scavenging efficiency of air pollutants, since they have been used to compare the precipitation removal of different pollutants in previous studies (Galloway et al., 1993; Guerzoni et al., 1995; Tuncel and Ungör, 1996; Shrestha et al., 2002; Hicks et al., 2005; Kulshrestha et al., 2009; Bourcier et al., 2012; Zhang et al., 2015). These studies demonstrated that scavenging ratios vary according to particle size distribution similar to the particle size dependency of scavenging coefficients typically used in wet deposition modeling (Wang et al., 2014). Thus, scavenging ratios of particulate-phase pollutants have been used as a surrogate for other particulate-phase pollutants with similar particle sizes (Cadle et al., 1990; Sakata and Asakura, 2007; Cheng et al., 2015).

The objectives of this study were to (1) analyze long-term geographical patterns and temporal trends in nitric acid and sulfur dioxide concentrations and the air concentrations and wet deposition of base cations and acidic anions; (2) examine geographical and temporal trends in aerosol acidity and acid rain; (3) determine scavenging ratios for particulate inorganic ions using precipitation and air concentrations; and (4) develop an approach for estimating particulate and gaseous species wet scavenging contributions to total nitrate, ammonium and sulfate wet deposition and their scavenging ratios.

## 2 Methodology

### 2.1 Data description

#### 2.1.1 CAPMoN datasets

24-hr integrated trace gas concentrations and particulate inorganic ion concentrations in air and precipitation are measured by CAPMoN at rural locations across Canada. This dataset was accessed online from the NAtChem database (Environment Canada, 2015a). The monitoring sites are located in various regions of Canada with the majority of the sites residing in the province of Ontario and Quebec (Fig. S1). Site elevations range from 14-707 m.a.s.l. The sites include continental and coastal sites, and the land use types of most CAPMoN sites are categorized as forested or agricultural sites (Table 1). Only the sites with long-term data (≥9 years) were analyzed in this study. The measurement periods are not synchronized between all sites; therefore, some sites may have different starting and end dates.

#### 2.1.2 Air concentrations



Non-size selective filters are used to collect air samples, which are analyzed for major inorganic ions ($Ca^{2+}$, $Mg^{2+}$, $K^+$, $Na^+$, $Cl^-$, $NH_4^+$, $NO_3^-$, and $SO_4^{2-}$) and $HNO_3$ and $SO_2$ trace gases. Teflon filters are used for the inorganic ions, while nylon and impregnated cellulose filters are used for $HNO_3$ and $SO_2$, respectively. The filters are placed in a three stage filter pack, which samples air at 10 m above ground. Every 24 hr at 8:00 LT, the sequential sampler passes air through a different filter

pack. The filters are retrieved and delivered to CAPMoN laboratories for chemical analysis. The collected mass for each species, blank values, and mass flow rates are used to determine the air concentrations in µg m$^{-3}$. Quality control of the data are performed using the Research Data Management and Quality Assurance System (RDMQ$^{TM}$) software (McMillan et al., 2000), which processes and manages large amounts of data, applies quality control checks, and assigns validity flags to each data point. The standard data flags warn users of missing values, invalidated values, valid values below detection limit, valid

estimated and interpolated values, non-conforming sampling periods, and valid values that have been replaced by the detection limit. In this study, valid air concentrations from 16 sites were analyzed. Available air measurements between 1983 and 2010 at the 16 sites are shown in Table 1.

### 2.1.3 Precipitation concentrations

Precipitation is sampled using a wet-only precipitation collector, which automatically opens when sensors detect

precipitation. Precipitation is collected in a specially-designed plastic bag in the collector. The bags containing precipitation are retrieved each day between 8:00 and 9:00 LT and are sealed, weighed and then kept refrigerated. The 24-hr integrated precipitation samples are analyzed for $H^+$ and major inorganic ions (described in Sect. 2.1.2) in concentrations of mg l$^{-1}$. Precipitation amount is also recorded daily at the precipitation monitoring sites for the determination of wet deposition flux. Standard data flags similar to the air data are applied to the precipitation data. In this study, valid precipitation

concentrations from 30 sites were analyzed. Available precipitation measurements between 1984 and 2011 at the 30 sites (two collocated collectors at Egbert) are shown in Table 1.

### 2.1.4 Meteorological data

Hourly air temperature, relative humidity and wind direction data at collocated or nearby stations to CAPMoN sites were obtained from the Canadian Climate Data Archives (Environment Canada, 2015b). Hourly air temperature and relative

humidity were averaged to daily values to correspond with the sampling intervals of the air and precipitation concentrations. Hourly wind direction data were used to determine the prevailing wind directions at each site.

### 2.2. Data analysis

### 2.2.1 Long-term patterns

Geographical patterns and temporal trends in air concentrations and wet deposition were examined. The geographical

patterns in air concentrations were based on 24-hr integrated measurements. For wet deposition, geographical patterns in the





annual wet deposition flux were analyzed since annual wet deposition is typically reported in previous studies. Daily precipitation concentrations were multiplied by the corresponding daily precipitation amount above the rain gauge detection limit of 0.2 mm. The annual wet deposition flux was obtained by summing the daily wet deposition flux. Thus, only the years with complete wet deposition data are used to determine the annual wet deposition. Statistical analyses of temporal

trends were performed using regression analysis of the annual average air concentrations and annual wet deposition for all years with complete data. The Mann-Kendall Test and Seasonal Kendall Test were also applied to the annual wet deposition and annual average air concentrations, respectively, to assess whether there was a statistically significant monotonic trend (Gilbert, 1987; Prestbo and Gay, 2009; Zbieranowski and Aherne, 2011; Cole et al., 2014). The Seasonal Kendall and Sen's estimator of slope provides the magnitude of the temporal trend on a per year basis. The Seasonal Kendall test is similar to

the Mann-Kendall test, except it analyzes the temporal trend in each month separately (Gilbert, 1987). The Mann-Kendall test was used to obtain the annual wet deposition trend because the magnitude of the slope (trend) based on monthly total wet deposition (if the Seasonal Kendall test was applied) is inconsistent with that of the annual wet deposition trend since wet deposition depends on the precipitation amount. The relationship between meteorological factors and temporal trends in air concentrations and wet deposition were also examined by correlation analysis.

**2.2.2 Aerosol acidity**

The molar cation/anion charge equivalent ratio (c/a) (Hennigan et al., 2015) was used as a measure of $H^+$ and aerosol acidity for the air sampling sites. The ions associated with acidic aerosols are predominantly $SO_4^{2-}$ and $NO_3^-$; however there are also contributions from organic acids which have not been accounted for by the c/a (Kerminen et al., 2001). Base cations and $NH_4^+$ are involved in neutralizing acidic aerosols. A c/a near 1 indicates that the aerosols are generally neutral, whereas a

lower ratio near 0.75 is indicative of acidic aerosols (Zhang et al., 2007; He et al., 2012). Daily c/a were determined only if all the ion measurements are available.

**2.2.3 Scavenging ratio**

In this study, monthly scavenging ratios (W) were first determined for ions existing only in the particulate phase ($Ca^{2+}$, $Mg^{2+}$, $K^+$, $Na^+$, and $Cl^-$) using Eq. 1 (Kasper-Giebl et al., 1998; He and Balasubramanian, 2008):

$$W = \frac{C_{prec}}{C_{air}} \times \frac{\rho_a}{\rho_w},\qquad(1)$$

$C_{prec}$ and $C_{air}$ are the precipitation and air concentrations, respectively. Surface air concentrations were used, even though in theory most of the scavenging occurs at the cloud height (Duce et al., 1991). $\rho_a$ and $\rho_w$ are the densities of air (1.2 kg m$^{-3}$) and water, respectively, which are used to convert the scavenging ratios to a mass basis. Scavenging ratios were determined on a monthly basis because they have less variability compared to daily (i.e. paired) or precipitation event scavenging ratios



(Galloway et al., 1993) and consider the average air concentration during both precipitation and dry periods (Kasper-Giebl et al., 1998). For the calculation of monthly scavenging ratios in Eq. 1, monthly volume-weighted precipitation concentrations and monthly average air concentrations based on ≥15 daily measurements in each month were used. Only daily precipitation concentrations with at least 0.2 mm precipitation amount were included. If there is insufficient data in each month, the scavenging ratio is not calculated. To account for the dependence on particle size distribution, the scavenging ratio of coarse PM ($W_{cPM}$) was determined by averaging $W_{Ca}$, $W_{Mg}$, and $W_{Na}$ since these base cations are predominantly in coarse PM (Cheng et al., 2015). $W_K$ was used as a surrogate for the scavenging ratio of fine PM ($W_{fPM}$) for inland sites, whereas $W_{K/2}$ was assumed for coastal sites following the methodology in Cheng et al. (2015). Scavenging ratios and wet scavenging in general are also affected by the chemical composition of aerosols, gas solubilities, temperature, precipitation amount and type, droplet size, nucleation efficiency, vertical concentration differences, and cloud type, which can contribute to the large variability in the scavenging ratios (Cadle et al., 1990; Duce et al., 1991; Galloway et al., 1993).

**2.2.4 Relative contributions and scavenging ratios of gaseous and particulate phases**

CAPMoN sites measure total $NO_3^-$, $SO_4^{2-}$, and $NH_4^+$ in precipitation. However, wet deposition of $NO_3^-$, $SO_4^{2-}$, and $NH_4^+$ can be attributed to the precipitation scavenging of particulates and gases, such as $pNO_3^-$ and $HNO_3$, $pSO_4^{2-}$ and $SO_2$, and $pNH_4^+$ and $NH_3$ (Kajino and Aikawa, 2015). To determine their relative contributions and the scavenging ratios of gases, particulate wet scavenging is first determined using $W_{cPM}$, $W_{fPM}$, particulate air concentration, and mass fractions in fine and coarse PM. The difference between the total wet scavenging and particulate wet scavenging is assumed to be due to the precipitation scavenging of gases. This assumption was also used in previous studies to estimate $NO_3^-$, $HNO_3$, $SO_4^{2-}$, and $SO_2$ scavenging ratios (Cadle et al., 1990) and the wet scavenging contributions by gaseous oxidized mercury and particulate mercury (Sakata and Asakura, 2007; Cheng et al., 2015). Eq. 2 was used to determine the wet scavenging of $pNO_3^-$:

$$[pNO_3^-]_{prec} = W_{fPM} [pNO_3^-]_{air} P_f + W_{cPM} [pNO_3^-]_{air} (1-P_f), \qquad (2)$$

$W_{fPM}$ and $W_{cPM}$ are the monthly scavenging ratios of fine and coarse PM, respectively (Sect. 2.2.3). $[pNO_3^-]_{air}$ is the monthly average $NO_3^-$ air concentration. $P_f$ is the mass fraction of $NO_3^-$ in fine PM, which varies with air mass origins and tends to form at lower temperatures (Zhang et al., 2008; Zhao and Gao, 2008). A $P_f$ of 0.84 was assumed for the winter months (DJF), whereas 0.29 was used for all other months. These are average mass fractions observed at CAPMoN sites in a short-term field study (Zhang et al., 2008). Eq. 2 accounts for the different scavenging efficiencies of small and large particles.

The contribution of $HNO_3$ to nitrate wet deposition was calculated using Eq. 3:

$$[HNO_3]_{prec} = [total \ NO_3^-]_{prec} - [pNO_3^-]_{prec,} \qquad (3)$$





[total $NO_3^-$]$_{prec}$ is the monthly volume-weighted $NO_3^-$ precipitation concentration and [$pNO_3^-$]$_{prec}$ is the wet scavenging of $pNO_3^-$ calculated from Eq. 2. If [$HNO_3$]$_{prec}$ < 0, it is assumed that only $pNO_3^-$ contributed to total nitrate precipitation and no gas scavenging occurred. The relative contributions of particulate and gaseous species to $NO_3^-$ wet deposition were determined using Eq. 4 and 5. Scavenging ratios of $pNO_3^-$ and $HNO_3$ were determined using Eq. 1.

$$\%pNO_3^- = ([pNO_3^-]_{prec}/ [\text{total } NO_3^-]_{prec})*100\%, \quad\quad (4)$$

$$\%HNO_3 = ([HNO_3]_{prec}/ [\text{total } NO_3^-]_{prec})*100\%, \quad\quad (5)$$

Similarly, Eq. 2-5 were used to determine the relative contributions of $pSO_4^{2-}$ and $SO_2$ to total $SO_4^{2-}$ wet deposition and $pNH_4^+$ and $NH_3$ to total $NH_4^+$ wet deposition. A $P_f$ of 0.94 for [$pSO_4^{2-}$]$_{air}$ and $P_f$ of 0.954 for [$pNH_4^+$]$_{air}$ was used for all months because these were the average mass fractions observed at CAPMoN sites (Zhang et al., 2008). $pSO_4^{2-}$ and $pNH_4^+$ have similar $P_f$ because they have similar particle size distributions and often exist together as $(NH_4)_2SO_4$ in the atmosphere. Scavenging ratios for $NH_3$ were not determined because $NH_3$ air concentrations were not available.

## 3 Results and Discussion

### 3.1 Air concentrations

#### 3.1.1 Geographical patterns

Air concentration statistics and geographical patterns of eight particulate inorganic ions, $SO_2$, and $HNO_3$ are plotted in Fig. 1a and b. The data were divided into two time periods from 1983-1996 and 1997-2010 to examine potential changes in concentrations due to $NO_x$ and $SO_2$ emission changes. The range in concentrations (based on the 5th percentile to 95th percentile concentration) from all daily samples at all locations was 0.009-2.9 µg m$^{-3}$ for $Ca^{2+}$, 0.002-0.5 µg m$^{-3}$ for $Mg^{2+}$, and 0.006-0.2 µg m$^{-3}$ for $K^+$. Larger variability was observed in $Ca^{2+}$ likely because of the variability of soil emissions depending on land use and wind. Large variability is also expected for $Na^+$ and $Cl^-$ because coastal sites are more frequently impacted by sea-salt aerosols than continental sites. The $Na^+$ and $Cl^-$ air concentrations ranged from 0.005-1.4 µg m$^{-3}$ and 0.003-1.9 µg m$^{-3}$, respectively. The range in concentrations were 0.018-5.8 µg m$^{-3}$ for $NH_4^+$, 0.009-8.7 µg m$^{-3}$ for $NO_3^-$, and 0.07-14.5 µg m$^{-3}$ for $SO_4^{2-}$. $HNO_3$ and $SO_2$ ranged from 0.014-5.0 µg m$^{-3}$ and 0.011-25.2 µg m$^{-3}$, respectively. These ions and trace gases are likely to have larger variability in air concentrations than base cations because some sites may be impacted more by anthropogenic emissions, which form secondary pollutants such as $SO_4^{2-}$, $HNO_3$, $NH_4^+$, and $NO_3^-$, than other sites.

The geographical patterns in air concentrations were examined in greater detail based on the median concentration at each location. Long-term median $Ca^{2+}$ concentrations among the sites ranged from 0.03-0.6 µg m$^{-3}$. The highest median during both time periods were observed at Longwoods and Egbert, which are the lowest latitude and most inland air concentration



sites. Longwoods and Egbert are also predominantly agriculture sites. The median $Mg^{2+}$ concentrations ranged from 0.01-0.09 µg m$^{-3}$. The highest median was also observed at Longwoods and Egbert. Higher median concentrations were also found at several western Canada sites including at Bratt's Lake, Esther and Saturna in the post-1997 period. The median concentrations ranged from 0.02-0.06 µg m$^{-3}$ for $K^+$, which is the ion with the least spatial variability in the air concentration.

The highest concentrations were observed at Longwoods as well as at Bratt's Lake post-1997. The median $Na^+$ and $Cl^-$ concentrations ranged from 0.02-0.5 µg m$^{-3}$ and 0.007-0.3 µg m$^{-3}$, respectively. As expected, the highest median for both ions were observed at the two coastal locations, Saturna and Kejimkujik, due to the proximity to sea-salt aerosol emissions from the ocean. These two sites are the farthest west and east air sampling locations respectively. $Na^+$ and $Cl^-$ concentrations at Saturna were larger and had greater variability than at Kejimkujik likely because of the higher frequency of marine airflows arriving at Saturna (68% of winds from N and W directions) than at Kejimkujik (31% of winds from E and S directions).

The median $NH_4^+$ and $NO_3^-$ concentrations ranged from 0.1-1.7 µg m$^{-3}$ and 0.03-2.0 µg m$^{-3}$, respectively (Fig. 1a and b). Compared to pre-1997 period, the median concentrations of $NH_4^+$ and $NO_3^-$ were lower in the post-1997 period. The highest concentrations for both ions were observed at Longwoods and Egbert. Higher concentrations were also found at Sutton, Esther, and at Frelighsburg post-1997. The majority of these sites except for Sutton are agriculture sites located in southern Ontario and Quebec, which implies that higher ammonia emissions from agricultural regions may react with acidic gases in the atmosphere to form particulate ammonium (Pitchford et al. 2009). Acidic gases, such as $H_2SO_4$ and $HNO_3$, are produced from the oxidation of $SO_2$ and $NO_x$ respectively and are primarily emitted from industrial and urban areas. The proximity of these lower latitude air sampling sites to major industrial areas in Ohio and Pennsylvania, USA could result in higher acidic gas concentrations at these sites. This is evident in the air concentration plots for $HNO_3$ that show higher concentrations of $HNO_3$ at sites having higher $NO_3^-$. The median $HNO_3$ concentrations ranged from 0.07-1.1 µg m$^{-3}$. Southerly winds also impacted Longwoods, Egbert, and Frelighsburg/Sutton approximately 20%, 32%, and 34% of the time, respectively. The median $SO_4^{2-}$ concentrations among the air sampling sites ranged from 0.6-3.5 µg m$^{-3}$. The concentrations were lower during the post-1997 than during the pre-1997 period. The highest median concentration was observed at Longwoods. Higher median concentrations were found at several southern Ontario and Quebec sites including Egbert, Sutton, Frelighsburg, Sprucedale, and Chalk River. Larger variability in the concentrations was generally observed across sites in southern and eastern Canada. This pattern is likely attributed to the proximity of the sites to combustion and industrial sources in southern Ontario and Quebec. The southern Canada and Atlantic Canada sites (e.g. Kejimkujik) are downwind of combustion and industrial areas in Ohio and Pennsylvania. In contrast, $SO_4^{2-}$ concentrations at sites located in western and central Canada (e.g., Saturna, Esther, Cree Lake, Bratt's Lake, and ELA) were at or below the overall median concentration of all the sites and had smaller variability.

The median $SO_2$ concentrations ranged from 0.4-6.4 µg m$^{-3}$ during the 1983-1996 period and from 0.6-2.3 µg m$^{-3}$ post-1997 (Fig. 1b). There was also a reduction in the variability of the concentrations in the post-1997 period. The lower latitude



southern Ontario and Quebec sites had higher $SO_2$ concentrations, whereas western and central Canada sites had much lower concentrations. This geographical pattern is similar to that of $SO_4^{2-}$. One exception was that the $SO_2$ concentrations in eastern Canada were similar to or even lower than in western Canada, whereas $SO_4^{2-}$ concentrations in eastern Canada were slightly higher than in western Canada. Eastern Canada sites including Montmorency, Lac Edouard, and Kejimkujik are

5 remote sites; therefore $SO_2$ concentrations are likely not elevated by the time it arrives at these remote locations since $SO_2$ can undergo deposition or transform to $SO_4^{2-}$ during transport. The slightly higher $SO_4^{2-}$ in eastern Canada could be from sea-salt sulfate due to the proximity to the Atlantic Ocean.

### 3.1.2 Temporal patterns

The Kendall slopes and confidence interval in Table 2 shows the annual rate of change in the concentrations of particulate

ions and trace gases at the air sampling sites for all years with available data. A significant temporal trend in $Ca^{2+}$ was observed at 5 of 16 sites with no significant changes in the concentrations observed at the remaining sites. Decreasing trends were observed at Saturna, Longwoods, and Egbert, while increasing trends were observed at Esther and Kejimkujik. The largest decline in $Ca^{2+}$ was -16 ng m$^{-3}$ a$^{-1}$ at Longwoods. Overall the temporal changes in $Ca^{2+}$ are small. For $Mg^{2+}$, a significant decreasing trend was found at Saturna, Longwoods, Chalk River, and Kejimkujik. However, the rate of decline

was very small ranging only from -0.5 to -1.1 ng m$^{-3}$ a$^{-1}$. The rate of decline for $K^+$ ranged from -0.5 to -3.8 ng m$^{-3}$ a$^{-1}$ and was observed at 14 of 16 sites. A plot of the annual average $K^+$ for ten of the active air sampling sites are shown in Fig. 2a for the 1983-2010 period, which illustrates a gradual decline in $K^+$. Significant decreases in $Na^+$ were found at 11 of 16 sites with magnitudes ranging from -0.3 to -4.5 ng m$^{-3}$ a$^{-1}$. The steepest decline was observed at a coastal site in western Canada. However, increasing trends were observed at two coastal sites in eastern Canada (Montmorency and Kejimkujik). For $Cl^-$,

decreasing temporal trends were found at only 6 of 16 sites, suggesting that the temporal trends are not necessarily related to sea-salt emissions. The decline in $Cl^-$, ranging from -0.1 to -0.8 ng m$^{-3}$ a$^{-1}$, were found at sites in western and central Canada and at Algoma and Longwoods.

$NH_4^+$ concentrations have been decreasing at 12 of 16 air sampling sites (Table 2). This result is consistent with the widespread decrease in $NH_4^+$ at CAPMoN sites during 1988-2007 (Zbieranowski and Aherne, 2011). The rate of decrease

ranged from -4 to -58 ng m$^{-3}$ a$^{-1}$. The largest declines as shown in Fig. 2b were observed at Longwoods, Egbert, Sprucedale, and Frelighsburg, which are agriculture sites located in southern Ontario and Quebec. The annual decrease was 7.3 times greater than other sites based on the linear regression slopes. The decreasing trend in $NH_4^+$ corresponds to the decreasing trend in ammonia emissions in Ontario and Quebec particularly in the post-2002 period (Fig. 2b) (Environment Canada, 2014). Aside from its relationship to ammonia, the negative trend in $NH_4^+$ was also strongly tied to trends in $NO_3^-$ and $SO_4^{2-}$.

There was an even split in the number of sites with increasing trends and decreasing trends in $NO_3^-$. The rate of increase ranged from 1.3-6.4 ng m$^{-3}$ a$^{-1}$ among active sites, whereas the annual trend was 6-40 ng m$^{-3}$ a$^{-1}$ among inactive sites (e.g. Esther, Sutton, Montmorency). The annual trend in $NO_3^-$ decreased from -9.3 to -53 ng m$^{-3}$ a$^{-1}$ at other sites (Table 2).





Larger declines were observed at the agriculture sites located in southern Ontario and Quebec. Differences in temporal trends were also observed during different time periods. At 9 of 16 sites, an increasing trend was found between 1991 and 2001 which was followed by a decreasing trend from 2001 to 2010 (Fig. 3). The difference in $NO_3^-$ trends between the two decades was also reported in previous analysis of CAPMoN sites (Zbieranowski and Aherne, 2011). The change in $NO_3^-$

temporal trends closely resembled that of $NO_x$ emissions in Canada. Between 1991 and 1997, $NO_x$ emissions in Canada increased annually and only began to decrease from 1997 to 2010 (Fig. 3) (Environment Canada, 2014). In the U.S., $NO_x$ emissions were constant over the 1991-1994 period and only began to decrease after 1994 (USEPA, 2015). Reductions in $NO_x$ emissions were implemented following the introduction of the Canada-U.S. Air Quality Agreement and the U.S. Acid Rain Program and Clean Air Interstate Rule. The decrease in $NO_x$ emissions were largely attributed to lower emissions from

stationary fuel combustion and transportation sectors (Lloret and Valiela, 2016).

$SO_4^{2-}$ decreased at a rate of -28 to -109 ng m$^{-3}$ a$^{-1}$ depending on the location (Table 2). The steepest annual declines were observed in the southern Ontario and Quebec region as shown in Fig. 4. The slope of the linear regression equation for the southern Ontario and Quebec sites in Fig. 4 was two times greater than that of other air sampling sites, which are coastal or higher latitude sites distant from major industrial and urban areas. The geographical patterns in the temporal trends of $SO_4^{2-}$

were also similar to those of $SO_2$ and $HNO_3$. The annual trends for $SO_2$ and $HNO_3$ in the southern Ontario and Quebec region declined 3.8 and 4.9 times faster, respectively, than other air sampling sites across Canada based on the linear regression slopes. Negative trends for $SO_4^{2-}$ and $SO_2$ concentrations followed the decreasing trend in $SO_2$ emissions in both Canada and U.S. since 1990 (Environment Canada, 2014; USEPA, 2015) (Fig. 4), corresponding to the period of the Canada-U.S. Air Quality Agreement and the U.S. Acid Rain Program and Clean Air Interstate Rule. Note the steeper

decline in $SO_2$ emissions in Ontario in recent years, which is potentially attributed to the phase-out of coal use in Ontario power plants beginning in 2005 (MOE, 2015).

The influence of meteorological parameters including temperature, relative humidity and precipitation rates on the temporal trends of particulate ions and trace gases were also investigated by performing correlation analyses on the monthly averaged data. The descending trend in $K^+$ between 1993 and 2010 at the majority of the sites was not strongly influenced by

precipitation as evident by the weak correlation coefficients (Table S1). Higher correlation between monthly average $K^+$ and temperature were found at Sprucedale, Frelighsburg and Lac Edouard (r = 0.52-0.69, p<0.05). At these sites, the monthly average $K^+$ peaked in March-April and was at the minimum concentration during December-January, which resembled the seasonal temperature cycle (Fig. S2a). The higher $K^+$ in the early spring could be attributed to increase soil emissions from agriculture operations and forest fires during springtime since the major sources of particulate $K^+$ are from biomass and soil.

Decreasing $NH_4^+$ observed at most of the sites was only weakly correlated with monthly precipitation rates and relative humidity, implying these meteorological parameters had little influence on the long-term temporal trend. Higher correlation between monthly average $NH_4^+$ and temperature was found at Kejimkujik (r = 0.63, p<0.05). The maximum $NH_4^+$ typically occurred during April-May and reached its lowest concentration during December-January (Fig. S2b). This seasonal trend is





linked to the formation of $SO_4^{2-}$ through $SO_2$ oxidation, which tends to occur at higher temperatures because of increase production of atmospheric oxidants. This theory is consistent with the very high correlation between monthly average $NH_4^+$ and $SO_4^{2-}$ (r = 0.91, p<0.05) at Kejimkujik. Overall, strong correlations between $NH_4^+$ and $SO_4^{2-}$ were observed at all the sites (r = 0.6-0.94, p<0.05). The high correlation between $SO_4^{2-}/SO_2$ ratio and temperature (r = 0.61-0.84, p<0.05) suggests

$SO_4^{2-}$ formation from the gas-phase oxidation of $SO_2$ (Yao et al., 2002). Besides the relationship between $NH_4^+$ and $SO_4^{2-}$, three of the sites including Saturna, ELA and Egbert exhibited strong correlations between $NH_4^+$ and $NO_3^-$ (r = 0.52-0.7, p<0.05) indicating that $NH_4^+$ also followed the temporal trend of $NO_3^-$ at some locations. In summary, precipitation and relative humidity had little impact on the long-term temporal patterns of particulate ions. Seasonal temperature trends is linked with the seasonal cycle of atmospheric oxidants which explains the short-term patterns in $SO_4^{2-}$ and $NH_4^+$; however,

there is no clear evidence that long-term temperature changes impacted their long-term trends.

### 3.1.3 Aerosol acidity (c/a)

The median c/a ranged from 0.97-1.6 and an overall median c/a of 1.07 was observed across all sites (Fig. 5a). These values are based on inorganic ion contributions to aerosol acidity. Organic acids do not contribute significantly to aerosol acidity compared to the strong inorganic acids (Zhang et al., 2007; Ziemba et al., 2007; He et al., 2012); however, they have been

included in the measure of aerosol acidity in some studies (Hennigan et al., 2015 and references therein). Among the sites, the highest aerosol acidity was observed at Kejimkujik because of the higher equivalent anion concentrations relative to cations. Higher aerosol acidity was prevalent generally in eastern Canada and central Ontario regions. In contrast to Kejimkujik, the majority of the sites had higher cation than anion concentrations or near equivalent cation and anion concentrations. Even though locations like Longwoods and Egbert had greater amount of anions than other locations due to

higher $NO_3^-$ and $SO_4^{2-}$, there were sufficient amounts of cations, mainly $NH_4^+$, to neutralize the acidic species. While the overall median c/a was close to 1 for all Canadian sites, more than half of the daily c/a data were below 1 at eight locations (Fig. 5a). This suggests there was a substantial amount of time between 1994 and 2010 when aerosols were acidic at some Canadian sites.

A significant increasing trend in the c/a was observed between 1994 and 2010, which indicates a widespread decline in

aerosol acidity (Fig. 5b). The rate of decrease in aerosol acidity was small and fairly uniform spatially based on the Kendall slope results. According to Fig. 5b, the annual average cation and anion concentrations and c/a at most of the sites (data combined from 13 of 15 sites) were relatively constant between 1994 and 2000. Since 2001, the annual average cation and anion concentrations and aerosol acidity have been on a slight decline. The decrease in cation concentrations appeared consistent with the declining trend in $NH_4^+$ discussed earlier, since $NH_4^+$ is the largest contributor to cation concentrations.

For anions, $NO_3^-$ and $SO_4^{2-}$ are the predominant ions. Thus, the decline in anions was also consistent with the decreasing rates for $NO_3^-$ and $SO_4^{2-}$. As mentioned earlier, the decreasing trends in $NH_4^+$, $NO_3^-$ and $SO_4^{2-}$ were consistent with the reductions in ammonia, $NO_x$ and $SO_2$ emissions.



### 3.2 Wet deposition

### 3.2.1 Geographical patterns

The annual wet deposition statistics for the various ions at the 30 locations are shown in Fig. 6. The annual wet deposition was based on all years with complete data because the annual flux was determined by summing the daily fluxes. The range

5   in the annual wet deposition based on the 5$^{th}$ and 95$^{th}$ percentile annual wet deposition rate (kg ha$^{-1}$ a$^{-1}$) was: 0.08-3.6 for Ca$^{2+}$, 0.02-1.6 for Mg$^{2+}$, 0.01-0.7 for K$^+$, 0.03-12.0 for Na$^+$, 0.06-23.0 for Cl$^-$, 0.1-6.4 for NH$_4^+$, 0.4-26.5 for NO$_3^-$, and 0.5-32.7 for SO$_4^{2-}$. The lowest annual wet deposition rates for Ca$^{2+}$, Mg$^{2+}$, Na$^+$, NH$_4^+$, NO$_3^-$, and SO$_4^{2-}$ were observed at Snare Rapids, which is a remote site in the Northwest Territories of Canada. The highest annual wet deposition recorded at the most eastern coastal site in Bay d'Espoir, Newfoundland were from ions related to sea-salt aerosols, including Na$^+$, Cl$^-$, and

10   Mg$^{2+}$. For Ca$^{2+}$ and ions derived from anthropogenic sources (e.g., NH$_4^+$, NO$_3^-$, and SO$_4^{2-}$), the highest annual wet deposition rates were observed at Priceville and Longwoods, which are the two most southern wet deposition sites in Canada and closest to urban and industrial areas.

Long-term median annual wet deposition of Ca$^{2+}$ ranged from 0.1-2.8 kg ha$^{-1}$ a$^{-1}$ among the wet deposition sites. The highest annual wet deposition was observed at Priceville and Longwoods (Fig. 6). Higher median wet deposition was also found at

Algoma, Egbert, and Warsaw Caves. The majority of these sites are agriculture sites. The lowest median wet deposition was observed at the western and eastern coastal sites. The median annual wet deposition of Mg$^{2+}$ ranged from 0.03-1.0 kg ha$^{-1}$ a$^{-1}$. The highest wet deposition was found at the western and eastern coastal sites with the exception of Goose Bay, which is a higher latitude coastal site located in Labrador. It had the lowest annual precipitation amount among coastal sites in eastern Canada (Fig. S3). Annual Mg$^{2+}$ wet deposition at inland sites was typically at or below the overall median wet

deposition for all sites; however, they are slightly higher at Longwoods and Priceville. The median annual wet deposition of K$^+$ ranged from 0.05-0.4 kg ha$^{-1}$ a$^{-1}$. Similar to Mg$^{2+}$, higher annual wet deposition of K$^+$ was observed at eastern coastal locations with the exception of Goose Bay. Annual wet deposition for inland sites was around the overall median annual wet deposition for all sites except at Longwoods, Algoma and Priceville. The median annual wet deposition of Na$^+$ and Cl$^-$ ranged from 0.05-7.5 kg ha$^{-1}$ a$^{-1}$ and 0.1-13.6 kg ha$^{-1}$ a$^{-1}$, respectively. The geographical patterns in the annual wet

deposition were similar between Na$^+$ and Cl$^-$. Annual wet deposition at western and eastern coastal sites were higher and had greater variability than inland locations.

Median annual NH$_4^+$ wet deposition ranged from 0.2-5.8 kg ha$^{-1}$ a$^{-1}$ (Fig. 6). Higher annual wet deposition was observed at lower latitude continental sites. Higher latitude continental locations (e.g. Cree Lake, Island Lake, Pickle Lake, Bonner Lake, Chapais) and coastal locations were well below the overall median annual wet deposition. The median annual wet

deposition ranged from 0.8-23.3 kg ha$^{-1}$ a$^{-1}$ for NO$_3^-$ and 0.8-26.6 kg ha$^{-1}$ a$^{-1}$ for SO$_4^{2-}$. These two ions have similar spatial patterns in annual wet deposition. Higher median annual wet deposition occurred at southern Ontario and Quebec sites. Lower median annual wet deposition was observed in eastern Canada. The lowest median annual wet deposition for NO$_3^-$





and $SO_4^{2-}$, which were well below the overall median annual wet deposition for all sites, was recorded in western and central Canada.

The geographical patterns in the wet deposition were predominantly affected by the air concentrations. For example, the higher $Na^+$ and $Cl^-$ wet deposition at coastal locations can be traced back to the higher $Na^+$ and $Cl^-$ air concentrations. Similarly, the higher $NH_4^+$, $NO_3^-$ and $SO_4^{2-}$ wet deposition occurring at southern Ontario and Quebec locations was consistent with the geographical patterns in the air concentrations. Although precipitation amount is used to determine wet deposition, only the wet deposition patterns of $Mg^{2+}$ and $K^+$ were potentially influenced by precipitation amount. As shown in Fig. S3, the annual precipitation amount generally increases from western to eastern sites. Only the $Mg^{2+}$ and $K^+$ wet deposition were higher at eastern Canada locations.

## 3.2.2 Temporal trends

Long-term temporal trends in the annual wet deposition of ions were analyzed using Sen's slope (Table 3 and 4) and linear regression analysis. The annual wet deposition of $Ca^{2+}$ and $K^+$ has not changed significantly at almost all locations. For $Mg^{2+}$and $Na^+$, there were no significant changes in the annual wet deposition rate at most of the sites, while a very small statistically significant decline in the annual wet deposition was observed at other sites. Of these sites, the rate of decline ranged from -0.001 to -0.008 kg ha$^{-1}$ a$^{-1}$ for $Mg^{2+}$ and -0.002 to -0.02 kg ha$^{-1}$ a$^{-1}$ for $Na^+$. Decline in base cations has been reported at other Canadian sites during the 1990s (Watmough et al., 2005). The small decrease or lack of change in base cation wet deposition is expected because the major source of base cations at rural Canadian sites are from natural emissions (Watmough et al., 2005). Monitoring the wet deposition $Ca^{2+}$, $Mg^{2+}$, $K^+$, and $Na^+$ trends are important because these ions neutralize soil acidity and mitigate further harmful impacts to plants and wildlife. A declining trend in the $Cl^-$ wet deposition was observed at 5 of 16 sites (Fig. 7a), and the magnitude ranged from -0.007 to -0.03 kg ha$^{-1}$ a$^{-1}$ (Table 3).

Significant trends in the annual wet deposition of $NH_4^+$ were observed at only 3 of 16 locations. There were two sites with increasing temporal trend (0.05 kg ha$^{-1}$ a$^{-1}$), whereas a decreasing trend was found at one location (Table 4). A lack of an overall consistent temporal trend in precipitation $NH_4^+$ was also found in previous analysis of CAPMoN sites (Zbieranowski and Aherne, 2011). These trends were in contrast to those in the U.S., which has seen an increase in precipitation $NH_4^+$ at 64% of the wet deposition sites between 1985 and 2004 (Lehmann et al., 2007). While increasing $NH_4^+$ in precipitation helps to increase precipitation pH and promote plant growth, a counter-effect is that soils can become acidic when $NH_4^+$ undergoes nitrification (Vogt et al., 2006).

Declining trends in $NO_3^-$ wet deposition was observed at 10 of 16 sites (data combined in Fig. 7a), while no significant trend was found at other locations. The rate of decrease ranged from -0.07 to -1.0 kg ha$^{-1}$ a$^{-1}$ and was largest at the southern Ontario and Quebec sites (Algoma, Longwoods, Egbert, Sprucedale, Frelighsburg, Lac Edouard). One exception was the non-significant trend at Sutton (Table 4) even though this site is only 15 km from Frelighsburg. The discrepancy in the



temporal trends at these two sites can be partially attributed to the different measurement periods. Measurements of wet deposition at Sutton ended in 2002, whereas Frelighsburg has been actively measuring wet deposition since 2001. This suggests the rate of decrease in $NO_3^-$ wet deposition was more rapid in the period after 2001 and corresponds with the decline in $NO_3^-$ air concentrations over the same time period discussed earlier. In the U.S. northeast, the decline in precipitation $NO_3^-$ was observed at only 25% of the sites in that region during 1985-2004 (Lehmann et al., 2005). A recent study of $NO_3^-$ wet deposition from 1985-2011 across North America indicates a 40-50% decrease in eastern North America after 2000 (Lloret and Valiela, 2016). Similar to the air concentrations of $SO_4^{2-}$, decline in $SO_4^{2-}$ wet deposition was also prevalent throughout Canadian sites. This finding is consistent with the decrease in precipitation $SO_4^{2-}$ reported at other Canadian sites during the 1990s (Watmough et al., 2005) and at 89% of the wet deposition sites in the U.S. between 1985 and 2004 (Lehmann et al., 2007). Decreasing temporal trends in $SO_4^{2-}$ wet deposition was found at 11 of 16 sites with magnitudes ranging from -0.1 to -1.0 kg ha$^{-1}$ a$^{-1}$ depending on the location (Table 4). The overall rate of decline was ~2 times higher at the southern Ontario and Quebec sites relative to other locations (Fig. 7b), which is consistent with the patterns in the air concentrations of $SO_4^{2-}$ described earlier. The large declining trends in the wet deposition of nitrogen and sulfur-containing species in conjunction with the relatively smaller declines or lack of change in base cations indicate that acid rain has attenuated over time. This is largely attributed to policies controlling $NO_x$ and $SO_2$ emissions.

Daily wet deposition of ions was correlated with their respective daily particulate matter concentrations and daily precipitation amount to gain insight into factors influencing the temporal trends in wet deposition. Moderate correlations (r = 0.38-0.41, p<0.05) between daily wet deposition and particulate matter concentration were found for $SO_4^{2-}$, $Na^+$ and $Cl^-$, whereas only weak correlations were found for other ions. This result partly explains the prevalent decline in wet deposition of $SO_4^{2-}$ and $Cl^-$. Decreasing $NO_3^-$ wet deposition was also widespread; however it did not strongly correlate with particulate $NO_3^-$ concentrations (r = 0.21, p<0.05). This is potentially because both gaseous and particulate nitrogen species can contribute to $NO_3^-$ wet deposition. This is supported by the slightly higher correlation (r = 0.33, p<0.05) between daily $NO_3^-$ wet deposition and $HNO_3$. For $SO_4^{2-}$ wet deposition which can be attributed to the precipitation scavenging of particulate $SO_4^{2-}$ and $SO_2$, only a weak correlation was found between daily $SO_4^{2-}$ wet deposition and $SO_2$ (r = 0.13, p<0.05). Further analysis on the relative contributions of gaseous and particulate species to $NO_3^-$ and $SO_4^{2-}$ wet deposition will be discussed in Sect. 3.4. Moderate correlations between daily wet deposition and daily precipitation were found for $NH_4^+$, $NO_3^-$ and $SO_4^{2-}$ (r =0.50-0.56, p<0.05), while weaker correlations were found for other ions. Some correlation was expected because wet deposition is determined from precipitation concentration and precipitation amount. On an annual basis, there has been no significant change to the annual precipitation amount at most of the sites, except for significant increases at Bratt's Lake, ELA, Chalk River, and Kejimkujik (Table 4). The lack of change to the annual precipitation is inconsistent with the decreasing trends in $SO_4^{2-}$, $NO_3^-$ and $Cl^-$ annual wet deposition found at the majority of the sites. Thus, long-term wet deposition of ions was not strongly influenced by long-term precipitation trends between 1983 and 2010.





### 3.2.3 Acid rain

The geographical patterns in acid rain as measured by precipitation pH are shown in Fig. 8. Precipitation pH is slightly acidic by nature due to the presence of carbonic acid formed by the dissolution of $CO_2$. A pH below 5 is considered to be acidic precipitation (Lehmann et al., 2007). According to Fig. 8, the median pH in daily precipitation samples across the sites ranged from 4.4-5.7. Between 1983 and 2011, acidic precipitation was observed in more than 50% of the daily precipitation samples at 19 of 31 or 61% of the sites. Acidic precipitation was prevalent in southern Ontario and some parts of eastern Canada, whereas pH was above 5 in western and central Canada. Similarly in regions close to southern Ontario and eastern Canada, higher occurrences of acid rain have been observed in the U.S. northeast region during the 1994-1996 and 2002-2004 periods (Lehmann et al., 2007). Acid rain has contributed to the acidification of soil and lakes (Stoddard et al., 1999; Driscoll et al., 2001; Clair et al., 2002; 2003; Jeffries et al., 2003; Watmough and Dillon, 2003). In the southern Ontario and eastern Canada region, the acid deposition effects are more concerning because the soil is slightly acidic naturally and shallow and the underlying bedrock provides insufficient acid buffering capacity (Clair et al., 2002; Watmough and Dillon, 2003). The geographical patterns in precipitation pH were consistent with those of aerosol acidity discussed earlier. The correlation between the median pH and aerosol acidity (as measured by c/a ratio) among 15 locations was 0.68, which suggests acidic particles partially contributed to acid rain at various Canadian sites.

Between 1983 and 2011, an increasing trend in pH was observed at 13 of 16 sites, while no significant change in pH was found at the remaining three sites (Table 4). The overall decline in acidic precipitation in Canada is consistent with the trends in the U.S. (Lehmann et al., 2007). The rate of increase in pH was slightly higher at southern Ontario and Quebec sites (Table 4). Recent studies indicate there has been a gradual improvement to soil and surface water conditions due to decreases in $NO_3^-$ and $SO_4^{2-}$ wet deposition; however, this recovery has been outpaced by the rate of decline in acidic wet deposition (Strock et al., 2014; Lawrence et al., 2015). The increasing temporal trends in pH can be partially attributed to aerosol acidity. A correlation of 0.29 between daily pH and c/a was found in this study. Based on the annual trend between the 1994 and 2010 period, the annual average pH and c/a (data combined at 13 of 15 sites) had very similar trends (Fig. 5b) and the correlation coefficient improved to 0.86.

### 3.3 Scavenging ratios

#### 3.3.1 General statistics and comparisons with literature

A summary of the monthly average scavenging ratio (W) (on a mass basis) statistics for the inorganic ions and trace gases are provided in Table S2. Monthly $W_{Ca}$ ranged from 120-14338. The minimum and maximum values were found at Egbert and Algoma, respectively. $W_{Mg}$ ranged from 131-11243. The lowest W was recorded at Chapais, while the highest W was recorded at Kejimkujik. $W_{Na}$ ranged from 76-12165. The lowest W was recorded at Chapais, while the highest W was recorded at Kejimkujik similar to $Mg^{2+}$. The $W_K$ ranged from 69-5565 and had lower values compared to $Ca^{2+}$, $Mg^{2+}$, and





Na$^+$ because K$^+$ is predominantly in fine particulate matter except for an additional coarse mode found at coastal locations (Zhang et al., 2008). For this reason, it is assumed that the W of fine PM (W$_{fPM}$) was equivalent to the W$_K$ for inland sites, while W$_{K/2}$ was assumed for coastal sites. The minimum and maximum W$_K$ were found at Montmorency and Kejimkujik, respectively. Since Ca$^{2+}$, Mg$^{2+}$, and Na$^+$ are mainly associated with coarse particulate matter, the average W of these ions was used as an estimate of the W of coarse particles (W$_{cPM}$). The range of W$_{cPM}$ was 83-12165. The minimum and maximum values were found at Chapais and Kejimkujik, respectively. The W$_{Cl}$ ranged from 210-35521. The minimum and maximum values were found at Chapais and Algoma, respectively. Compared to other ions, W$_{Cl}$ were larger and had greater variability. Overall, the range in the average W for these ions among the 13 sites was within the average W from previous studies (Table S3).

Monthly scavenging ratios were determined for particulate NO$_3^-$ (pNO$_3^-$) and HNO$_3$ separately because both gaseous and particulate forms can contribute to NO$_3^-$ wet deposition. Monthly W ranged from 135-4272 for pNO$_3^-$ and 7-16658 for HNO$_3$. Based on the average scavenging ratio, W$_{HNO3}$ were greater than W$_{pNO3}$. The average W$_{pNO3}$ from 13 sites were within the range of literature values in Table S3; however the majority of the W$_{pNO3}$ in the literature are determined from total nitrate in precipitation and pNO$_3^-$ in air. Thus, most of the W$_{pNO3}$ are overestimated. Scavenging ratios of pNO$_3^-$ based on total nitrate in precipitation are higher by a factor of 1.4-18 depending on the site (average: factor of 6).

In this study, the average W$_{HNO3}$ at some sites were higher than those in Table S3; however, they were most comparable to the average determined by Cadle et al. (1990) likely because of the similarity in the methods of determining W$_{HNO3}$. The method first calculates the pNO$_3^-$ scavenged and then the difference between the total NO$_3^-$ and the pNO$_3^-$ scavenged is assumed to be contributed by HNO$_3$ scavenging. One major difference in the approach was that Cadle et al. (1990) used W$_K$ as a surrogate for W$_{pNO3}$ to estimate the pNO$_3^-$ scavenged, whereas in this study W$_{pNO3}$ considered the seasonal particle size distribution of pNO$_3^-$. The average W$_{HNO3}$ at some sites in this study were different than those determined by Hicks (2005) (Table S3), who assumed only HNO$_3$ contributed to NO$_3^-$ wet deposition. The values were also different from that of Kasper-Giebl et al. (1998), who used a multiple linear regression (MLR) approach. For comparison purposes, W$_{pNO3}$ and W$_{HNO3}$ derived from MLR are also shown in Table S3 and S4. The MLR results show W$_{HNO3}$>W$_{pNO3}$. However, this empirical method generated higher W$_{pNO3}$ and lower W$_{HNO3}$ at some locations compared to the method used in this study. Table S4 indicates that the MLR model fit was considered weak to moderate (R$^2$= 0.15-0.34) depending on the site.

Monthly W$_{pNH4}$ ranged from 63-4356. The lowest W was recorded at Egbert, while the highest W was recorded at Algoma. Scavenging ratios of NH$_3$ were undetermined because NH$_3$ air concentrations were not available. Average W$_{pNH4}$ were also lower than some of the values in the literature (Table S3), which are likely overestimated because the values were based on the total NH$_4^+$ precipitation concentrations, instead of pNH$_4^+$ scavenged by precipitation. Comparison of these two methods of calculation indicates that the use of total NH$_4^+$ precipitation concentration overestimated the scavenging ratios by 4-48% (average: 22%) depending on the location. Despite the coexistence of NH$_4^+$ and SO$_4^{2-}$ in the atmosphere, the difference




between the average scavenging ratios of these ions can vary by 4-98% (average: 32%). The range of monthly scavenging ratios for $pSO_4^{2-}$ and $SO_2$ were 75-3146 and 0.3-12068, respectively. The average $W_{pSO4}$ among the sites in this study were lower than some of the literature values in Table S3. Most of the studies excluded the wet scavenging of $SO_2$ because $pSO_4^{2-}$ was assumed to be the dominant contributor to $SO_4^{2-}$ wet deposition. This method overestimates the scavenging ratio of

$pSO_4^{2-}$ by 18-85% (average: 44%) compared to the method used in this study. According to the limited number of $W_{SO2}$ estimates (Table S3), the precipitation scavenging of $SO_2$ is less important compared to $pSO_4^{2-}$ because of the lower $W_{SO2}$. In this study, $SO_2$ and $pSO_4^{2-}$ can be equally important at times in terms of scavenging ratios. The average $W_{SO2}$ at more than half of the sites in this study were greater than literature averages. This is potentially due to the different methodologies for calculating $W_{SO2}$ and precipitation type. The MLR method yielded higher $W_{pSO4}$ and lower $W_{SO2}$ at some locations compared

to the method used in this study (Table S3 and S4). The approach used in this study was similar to Cadle et al. (1990); however, that study used $W_{NH4}$ as a surrogate for $W_{pSO4}$ based on the assumption these ions are typically found in the same aerosols. There are however large uncertainties with $W_{NH4}$ because of the scavenging by both $pNH_4^+$ and $NH_3$. As mentioned earlier, most of the $W_{NH4}$ in literature are overestimated because of the exclusion of $NH_3$. Use of these values could lead to a high bias in $W_{pSO4}$ and subsequently lower $W_{SO2}$. Aside from the methodology, the scavenging ratios

determined by Cadle et al. (1990) were based on snowfall events, which favor the scavenging of particles over gases (Hicks, 2005; Zhang et al., 2013; 2015).

The variability in scavenging ratios among different ions and trace gases within the same month is shown in Fig. S4. The range in scavenging ratios can be quite large. This is expected because the different physical and chemical properties of the pollutants affect their wet scavenging efficiencies. For particulate matter, coarse particles (e.g., $Ca^{2+}$, $Mg^{2+}$ and $Na^+$) are

scavenged more efficiently than fine particles (e.g. $K^+$) (Galloway et al., 1993; Guerzoni et al., 1995; Tuncel and Ungör, 1996). The different solubilities of gaseous pollutants ($HNO_3$ and $SO_2$) can also explain the differences in scavenging ratios for different pollutants. $W_{HNO3}$ were 1.4 to 6 times higher than those of $W_{SO2}$ depending on the location (Table S2), which is consistent with the higher solubility of $HNO_3$ compared to $SO_2$ (H = 2.1 x $10^5$ M atm$^{-1}$ vs. 1.2 M atm$^{-1}$; Zhang et al., 2006; Sander, 2015).

**3.3.2 Patterns in scavenging ratios**

Although the monthly scavenging ratios of most ions span a wide range, the average scavenging ratios of particulate ions were within a factor of 1.6-4.5 among the 13 sites (Table S2). The average scavenging ratios of $Na^+$, $Cl^-$ and $Ca^{2+}$ have larger spatial variability than other particulate ions. This may reflect the different precipitation scavenging efficiencies of particles in the marine boundary layer and continental atmospheres as hypothesized by Galloway et al. (1993). One related

theory is that the higher relative humidity in marine environments is conducive to the hygroscopic growth of sea-salt aerosols which increases its scavenging efficiency (Hennigan et al., 2008). The average scavenging ratios ranged from 497-996 for fine particles (factor of 2 spatial variability) and 666-2077 for coarse particles (factor of 3 spatial variability). Larger



scavenging ratios of fine particles were found at inland sites, where soil and biomass emissions are sources of $K^+$. For coarse particles, the larger scavenging ratios at coastal sites (Table S2) were likely attributed to oceanic source of $Na^+$ and $Mg^{2+}$. The greatest spatial variability was in the scavenging ratios of gases including $HNO_3$ and $SO_2$. The average scavenging ratios varied by a factor 7.4 and 10.7, respectively. The large variability in scavenging ratios between sites is expected

because the air and precipitation concentrations and precipitation type and amounts among other factors can also vary with location.

A pronounced seasonal variation can be seen in the monthly average scavenging ratios. The average scavenging ratio of most of the ions and $HNO_3$ were lowest during July or August (Fig. S5), which resembles the monthly $NO_3^-$, $SO_4^{2-}$, and $NH_4^+$ trends in a previous study (Kasper-Giebl et al., 1998). There were two periods when the average scavenging ratios peaked:

one peak during April-May and a second peak during September-October were observed for most of the ions and $HNO_3$. A similar pattern was obtained for $W_{pNO3}$ when MLR was used to generate monthly scavenging ratios. However, different patterns were obtained for $W_{HNO3}$ and $W_{pSO4}$ derived from MLR, as shown by the higher values during winter and lower values in the warm seasons (Fig. S5 and Table S5).

Monthly patterns in $W_{SO2}$ were different from those of other ions and $HNO_3$. The average $W_{SO2}$ were lower during winter

and peaked in the summer, and is supported by the MLR results as well (Fig. S5 and Table S5). This result is also consistent with the seasonal $W_{SO2}$ patterns from multiple U.S. sites (Hicks, 2005) and other studies suggesting the inefficient scavenging of $SO_2$ by snow (Kasper-Giebl et al., 1998 and references therein). However, limited field measurements of dissolved $SO_2$ (measured as sulfite) in precipitation samples indicate that the highest precipitation concentrations were found in the colder months, which is consistent with solubility theory (Hales and Dana, 1979; Dana, 1980). This finding does not

necessarily contradict the results from this study because $W_{SO2}$ also depend on the air concentration. The higher ambient $SO_2$ likely due to higher combustion emissions associated with winter heating (Fig. S6) resulted in lower scavenging ratios, and vice-versa during warmer months. Besides temperature effects on solubility, precipitation pH and the presence of $NH_3$ and $H_2O_2$ could also affect $SO_2$ wet scavenging (Zhang et al., 2006).

**3.4 Relative contributions of particulates and gases to nitrate, ammonium and sulfate wet deposition**

In the previous section, scavenging ratios were determined for particulate ($pNO_3^-$, $pNH_4^+$, $pSO_4^{2-}$) and gaseous ($HNO_3$ and $SO_2$) species. In this section, the relative percent contributions of these particulate and gaseous species to nitrate, ammonium, and sulfate wet deposition are determined. The average $\pm 1\sigma$ of $pNO_3^-$ contributions to nitrate wet scavenging (%$pNO_3^-$) was 28±23% for all years of data at the 13 locations. Percent $HNO_3$ contributions to nitrate wet scavenging (%$HNO_3$) was 72±23%. Based on the average, %$HNO_3$ dominated %$pNO_3^-$ at most of the sites (Fig. 9a). Geographical

variations were observed in the %$pNO_3^-$ and %$HNO_3$. Average %$pNO_3^-$ were higher at the two lowest latitude sites and coastal sites (Fig. 9a). One reason is because of the higher $pNO_3^-$ air concentrations at the lower latitude locations (Longwoods and Egbert) discussed in Sect. 3.1.1. %$pNO_3^-$ were also higher at the coastal locations (Saturna and



Kejimkujik) likely because of the partitioning of $HNO_3$ to sea-salt aerosols (Pryor and Sørensen, 2000; Fischer et al., 2006), which are typically coarse particles and hygroscopic and hence more efficiently removed by precipitation. In contrast, %$HNO_3$ were greater at the higher latitude continental locations (Fig. 9a). Relative contributions of $pNH_4^+$ and $NH_3$ were 70±19% and 30±19%, respectively. Precipitation scavenging of $pNH_4^+$ was greater than $NH_3$ at all sites (Fig. 9b). The

5   percent $pSO_4^{2-}$ contributions to sulfate wet scavenging (%$pSO_4^{2-}$) was 63±20%, while percent $SO_2$ contributions to sulfate wet scavenging (%$SO_2$) was 37±20%. Average %$pSO_4^{2-}$ were greater than that of %$SO_2$ at most of the sites (Fig. 9c). No pronounced geographical patterns were observed in the relative contributions of gases and particulates to ammonium or sulfate wet deposition. Knowledge of the relative scavenging contributions of gases and particles may improve the wet deposition modeling of nitrate, ammonium, and sulfate, which continues to show discrepancies between model and

observations (Appel et al., 2011; Zhang et al., 2012; Kajino and Aikawa, 2015; Qiao et al., 2015). Furthermore, gas- and particulate-phase pollutants may not come from the same source, since particulate matter can be re-emitted from natural sources and human activity. Aerosol formation is modeled separately in chemical transport models and the wet deposition of particles and gases use different parameterizations in these models. The scavenging coefficient of gases in models depends on Henry's law constant or gas diffusivity and reactivity, whereas the scavenging coefficient of particles is a function of

particle size distribution and collection efficiency among other factors (Gong et al., 2011). Studies also suggest different efficiencies between rainout and washout scavenging mechanisms for gases and aerosols (Gong et al., 2011; Kajino and Aikawa, 2015). These parameterizations can differ between different chemical transport models as well leading to large uncertainties in the wet deposition estimates (Tost et al., 2007). Gas/aerosol wet scavenging observations can also be used to evaluate those from wet deposition simulations (Kajino and Aikawa, 2015).

A seasonal cycle was observed in the relative contributions of gases and particulates to nitrate, ammonium, and sulfate wet deposition. The contributions by particulates to nitrate, ammonium, and sulfate wet scavenging were greater during cold months and lower during summer (Fig. 10). This pattern is consistent with studies suggesting that particle scavenging by snow is more efficient than the scavenging by rain for an equivalent amount of precipitation (Zhang et al., 2013, 2015). Based on scavenging ratios, Zhang et al. (2015) found that snow scavenging can be 10 times more efficient than the rain

scavenging of polycyclic aromatic compounds (PAC), and that snow scavenging of particulate-phase PAC can exceed that of gas-phase PAC by a similar magnitude. In contrast to particle scavenging, greater precipitation scavenging of gases was observed in the warm seasons (Fig. 10). This inverse relationship between particle and gas wet scavenging resulted because of Eq. 3, which assumes that the precipitation scavenging in excess of particle wet scavenging was due to gas scavenging. This assumption needs to be validated by independently deriving the gas scavenging contributions; however, the results

based on the assumption are consistent with precipitation scavenging theories.

In terms of nitrate scavenging, the largest difference between gas and particulate wet scavenging were observed during warm months (factor of 4 higher for $HNO_3$), whereas smaller differences were seen during cold months (Fig. 10a). The large contribution by $HNO_3$ is expected because it is one of the most soluble gases and is effectively scavenged by rainout (Chang,



1984; Garrett et al., 2006). %HNO$_3$ were also fairly high during the cold months because of higher solubility at lower temperatures and the high absorption and retention of HNO$_3$ and other strong acids on ice crystals (Diehl et al., 1995; Clegg and Abbatt, 2001). Snow scavenging of HNO$_3$ can also exceed below-cloud rain scavenging of HNO$_3$ for an equivalent precipitation rate (Chang, 1984).

Particle wet scavenging exceeded gas scavenging contributions to ammonium wet deposition in most months by a factor of 2.6 except during May-June (Fig. 10b). For sulfate scavenging, larger differences between gas and particulate wet scavenging were found during cold months (factor of 2 higher for pSO$_4^-$), while small differences were observed during warm months (Fig. 10c). The greater scavenging of pSO$_4^-$ during colder months is likely attributed to the effectiveness of particle scavenging by snow as discussed earlier. Another explanation for the large disparity between particle and gas
scavenging in the cold months could be the low absorption of SO$_2$ by ice crystals especially on low pH ice surfaces (Clegg and Abbatt, 2001). As for the smaller difference between particle and gas scavenging during warm months, experiments suggest that snow scavenging of SO$_2$ can be increased by the presence of H$_2$O$_2$ and relatively higher temperatures of ~0$^{\circ}$C (Mitra et al., 1990). The latter conditions are conducive to the formation of a quasi-liquid layer on the ice surface, which may increase the dissolution of SO$_2$ (Clegg and Abbatt, 2001).

**4 Conclusions**

Long-term air concentrations, wet deposition, and scavenging ratios of inorganic ions were analyzed using CAPMoN data. Geographical variability in the air concentrations of inorganic ions can be attributed to proximity of the sites to anthropogenic sources, oceanic sea-salt emissions, and agricultural emissions. Annual wet deposition geographical patterns for Ca$^{2+}$, Na$^+$, Cl$^-$, NH$_4^+$, NO$_3^-$, and SO$_4^{2-}$ were similar to those in air. Widespread declines were observed for NH$_4^+$ (1994-
2010) and SO$_4^{2-}$ (1983-2010) in air, which was attributed to decreases in SO$_2$, NO$_x$, and local NH$_3$ emissions. NO$_3^-$ air concentrations increased from 1991-2001 and then decreased from 2001-2010, consistent with the trends in NO$_x$ emissions in Canada over these two decades and in the U.S. over the last decade. However, widespread declines in annual wet deposition were only found for NO$_3^-$ and SO$_4^{2-}$ from 1984-2011. SO$_4^{2-}$ air concentrations and annual wet deposition declined ~2 times faster in southern Ontario and southern Quebec than other locations because of the proximity of the sites to
industrial emission sources. Aerosol acidity and acid rain had greater impacts to southern and eastern Canada than western Canada. Temporal trends show aerosol acidity and acid rain have been decreasing simultaneously from 1994-2010, consistent with large declines in nitrate and sulfur species and slight declines or lack of change in base cations.

Scavenging ratios of particulate NH$_4^+$, SO$_4^{2-}$ and NO$_3^-$ in literature may be overestimated on average by 22%, 44% and a factor of 6, respectively, because the wet scavenging of gases were excluded. The wet scavenging of HNO$_3$ dominated
particulate NO$_3^-$ at most locations, while the wet scavenging of particulate NH$_4^+$ and SO$_4^{2-}$ were more efficient than NH$_3$ and





$SO_2$, respectively. The wet scavenging of particles was more efficient in the cold months likely because of the scavenging by snow. Greater gas scavenging was found in the warm months opposite in trend to particulate wet scavenging.

Long-term trends in inorganic ions provide greater insight into which Canadian regions are still susceptible to or likely recovering from acid rain impacts, and the effectiveness of environmental policies at mitigating acid rain. Particulate

inorganic ions and trace gas scavenging ratios provide a measure of the wet scavenging efficiencies and are potentially useful surrogates for the wet scavenging of other pollutants provided that they have similar physicochemical properties (e.g. solubility, particle sizes, etc.). These results can be considered in future wet deposition modeling to improve the prediction of nitrate, ammonium, and sulfate wet deposition. Scavenging ratios can potentially be used to obtain a rough first order estimate of the wet deposition at other locations considering the uncertainties in both the scavenging ratios and in wet

deposition modeling.

**Data availability**

The datasets used in this study can be accessed from the websites in the reference list or by contacting the corresponding author.

**Competing interests**

The authors declare that they have no conflict of interest.

**Acknowledgements**

The authors gratefully acknowledge the CAPMoN team of Environment and Climate Change Canada (ECCC) and the Canadian National Atmospheric Chemistry (NAtChem) Particulate Matter and Precipitation Databases and its data contributing agencies/organizations for the provision of data (1983-2011) used in this publication. The authors thank

Chantale Cerny, Samantha Lee, and Bill Sukloff from ECCC for advice on climate data extraction.

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




Table 1: Site and data descriptions. NA indicates no available data. Short-term data were not analyzed. Refer to Fig. S1 for a map of the sites.

| Site name | Province | Latitude | Longitude | Elevation (m) | Coastal/inland | Land use | Air data | Wet deposition data |
|---|---|---|---|---|---|---|---|---|
| Saturna | BC | 48.78 | -123.13 | 178 | Coastal | Forest | Dec 1990-Dec 2010 | Jan 1990-Dec 2011 |
| Snare Rapids | NT | 63.52 | -116.00 | 240 | Inland | Forest | NA | Jan 1989-Dec 2011 |
| Esther | AB | 51.67 | -110.20 | 707 | Inland | Agricultural | Oct 1991-Mar 2003 | Jan 1987-Dec 2002, Jan 2009-Dec 2011 |
| Cree Lake | SK | 57.35 | -107.13 | 499 | Inland | Forest | Jul 1982-May 1993 | Jan 1984-Dec 1992 |
| Bratt's Lake | SK | 50.20 | -104.71 | 600 | Inland | NA | Aug 2001-Dec 2010 | Jan 2001-Dec 2011 |
| McCreary | MB | 50.71 | -99.53 | 335 | Inland | Agricultural | NA | Jan 1984-Dec 1995 |
| Island Lake | MB | 53.87 | -94.67 | 245 | Inland | Forest | NA | Jan 1984-Dec 1997 |
| Experimental Lakes Area (ELA) | ON | 49.66 | -93.72 | 369 | Inland | Forest | Jan 1979-Dec 2010 | Jan 1984-Dec 2011 |
| Pickle Lake B | ON | 51.45 | -90.22 | 370 | Inland | Forest | NA | Jan 2003-Dec 2011 |
| Algoma | ON | 47.04 | -84.38 | 411 | Inland | Forest | Oct 1980-Dec 2010 | Jan 1985-Dec 2011 |
| Burnt Island | ON | 45.82 | -82.95 | 185 | Inland | Forest | NA | Jan 1992-Dec 2011 |
| Bonner Lake | ON | 49.39 | -82.12 | 245 | Inland | Forest | short-term | Jun 1985-Dec 2011 |
| Longwoods | ON | 42.88 | -81.48 | 239 | Inland | Agricultural | Jan 1983-Dec 2010 | Jan 1984-Dec 2011 |
| Priceville | ON | 44.17 | -80.66 | 475 | Inland | Agricultural | NA | Jan 1985-Dec 1994 |
| Egbert | ON | 44.23 | -79.78 | 253 | Inland | Agricultural | Jul 1988-Dec 2010 | Jan 1989-Dec 2011 |
| Egbert-2 | ON | 44.23 | -79.78 | 253 | Inland | Agricultural | short-term | Jan 1997-Dec 2011 |
| Sprucedale | ON | 45.42 | -79.49 | 350 | Inland | Agricultural | May 2002-Dec 2010 | Jan 2003-Dec 2011 |
| Warsaw Caves | ON | 44.46 | -78.13 | 230 | Inland | Agricultural | NA | Jan 1986-Dec 2011 |
| Chalk River | ON | 46.06 | -77.41 | 184 | Inland | Forest | Jan 1979-Dec 2010 | Jan 1984-Dec 2011 |
| Chapais | QC | 49.82 | -74.98 | 381 | Inland | Forest | Jun 1988-Dec 2010 | Jan 1988-Dec 2011 |
| Frelighsburg | QC | 45.05 | -72.86 | 203 | Inland | Agricultural | Nov 2001-Dec 2010 | Jan 2002-Dec 2011 |
| Sutton | QC | 45.08 | -72.68 | 243 | Inland | Forest | Jan 1986-Mar 2002 | Jan 1984-Dec 2001 |
| Lac Edouard | QC | 47.68 | -72.44 | 243 | Inland | Forest | Jan 2002-Dec 2010 | Jan 2002-Dec 2011 |
| Montmorency | QC | 47.32 | -71.15 | 640 | Coastal | Forest | Dec 1980-Jan 1997 | Jan 1984-Dec 1996 |
| Harcourt | NB | 46.50 | -65.27 | 37 | Coastal | Forest | NA | Jan 1984-Dec 2011 |
| Kejimkujik | NS | 44.43 | -65.21 | 127 | Coastal | Forest | May 1979-Dec 2010 | Jan 1984-Dec 2011 |
| Mingan | QC | 50.27 | -64.22 | 14 | Coastal | Forest | short-term | Jan 1994-Dec 2011 |
| Jackson | NS | 45.59 | -63.84 | 90 | Coastal | Forest | NA | Jan 1984-Dec 2011 |
| Goose Bay | NL | 53.31 | -60.36 | 39 | Coastal | Forest | NA | Jan 1984-Dec 2011 |
| Goose Bay B | NL | 53.29 | -60.39 | 39 | Coastal | Forest | NA | Jan 1989-Dec 2007 |
| Bay d'Espoir | NL | 47.99 | -55.81 | 190 | Coastal | Forest | short-term | Jan 1984-Dec 2011 |



Table 2: Rate of change in annual air concentrations. Slope refers to the Seasonal Kendall slope (ng m$^{-3}$ a$^{-1}$); C.I. refers to the 90% confidence interval of the slope; ns indicates no significant trend; na indicates no available data.

| Site | Ca$^{2+}$ | | Mg$^{2+}$ | | K$^+$ | | Na$^+$ | | Cl$^-$ | |
|---|---|---|---|---|---|---|---|---|---|---|
| | Slope | C.I. | Slope | C.I. | Slope | C.I. | Slope | C.I. | Slope | C.I. |
| Saturna | -0.9 | -1.3 to -0.6 | -0.8 | -1.1 to -0.4 | -0.7 | -1 to -0.5 | -4.5 | -6.5 to -1.8 | 1.0 | ns |
| Esther | 9.9 | 4.8 to 15.8 | 1.5 | 0.9 to 2.5 | 1.3 | 0.8 to 1.8 | 2.1 | 0.8 to 3.6 | -0.4 | -0.7 to 0 |
| Cree Lake | na | na | na | na | -1.3 | -2 to -0.5 | 0.5 | ns | -0.5 | -1 to -0.1 |
| Bratt's Lake | 5.5 | ns | 2.4 | ns | -3.8 | -5.3 to -2.4 | -2.4 | -4.2 to -1.3 | -0.6 | -1.3 to -0.2 |
| ELA | 0.5 | ns | -0.2 | ns | -0.5 | -0.6 to -0.3 | -0.7 | -0.8 to -0.5 | -0.2 | -0.4 to -0.2 |
| Algoma | -0.3 | ns | -0.2 | -0.5 to 0 | -0.6 | -0.8 to -0.5 | -0.3 | -0.5 to -0.2 | -0.1 | -0.2 to 0 |
| Longwoods | -15.9 | -21.1 to -10 | -1.1 | -2.1 to -0.4 | -0.6 | -0.9 to -0.3 | -0.5 | -0.8 to -0.2 | -0.8 | -1 to -0.5 |
| Egbert | -9.7 | -14.8 to -3.6 | 0.1 | ns | 0.02 | ns | -0.2 | ns | 0.2 | ns |
| Sprucedale | -2.7 | ns | -0.9 | ns | -1.2 | -1.8 to -0.6 | -1.3 | -1.9 to -0.4 | -0.2 | ns |
| Chalk River | -0.3 | ns | -0.6 | -0.8 to -0.4 | -0.8 | -1 to -0.6 | -0.5 | -0.8 to -0.3 | 0.03 | ns |
| Chapais | 0.4 | ns | -0.03 | ns | -0.5 | -0.6 to -0.4 | -0.9 | -1.2 to -0.7 | 0.3 | 0 to 0.6 |
| Frelighsburg | -0.8 | ns | 0.3 | ns | -1.1 | -1.8 to -0.7 | -2.6 | -4.4 to -0.7 | -0.01 | ns |
| Sutton | 1.5 | ns | 0.2 | ns | -1.8 | -2.3 to -1.4 | -1.8 | -2.3 to -1.4 | -0.2 | ns |
| Lac Edouard | -0.01 | ns | -0.2 | ns | -1.0 | -1.4 to -0.6 | -1.4 | -2.1 to -0.5 | -0.4 | ns |
| Montmorency | na | na | na | na | -0.6 | -1.1 to 0 | 1.4 | 0.8 to 2.1 | 0.1 | ns |
| Kejimkujik | 0.4 | 0 to 0.8 | -0.5 | -0.8 to -0.1 | -0.5 | -0.7 to -0.4 | 1.8 | 0.4 to 3 | 4.9 | 3.7 to 6.6 |

| Site | NH$_4^+$ | | NO$_3^-$ | | SO$_4^{2-}$ | | SO$_2$ | | HNO$_3$ | |
|---|---|---|---|---|---|---|---|---|---|---|
| | Slope | C.I. | Slope | C.I. | Slope | C.I. | Slope | C.I. | Slope | C.I. |
| Saturna | -7.6 | -9.1 to -6.2 | -9.3 | -12.8 to -5.6 | -28.8 | -32.6 to -25.2 | -62.6 | -72.5 to -52.5 | -17.1 | -20.7 to -13.6 |
| Esther | 7.9 | 2 to 15.6 | 40.1 | 32.6 to 50.4 | 7.0 | ns | -6.5 | ns | 12.0 | 2.2 to 21.8 |
| Cree Lake | 3.7 | 1.1 to 6.2 | -1.1 | ns | 0.0 | ns | 8.4 | ns | 8.9 | 6 to 11.4 |
| Bratt's Lake | -16.5 | -23.4 to -9.7 | -21.4 | -34.3 to -10.5 | -30.4 | -51.1 to -11.8 | 1.6 | ns | -19.4 | -27.3 to -13.3 |
| ELA | -4.4 | -6 to -2.6 | 3.5 | 1.7 to 5.2 | -28.4 | -32.1 to -24.4 | -14.6 | -17.7 to -12.2 | -2.3 | -3.7 to -0.8 |
| Algoma | -7.0 | -9.9 to -3.7 | 6.4 | 4.4 to 8.4 | -46.8 | -55 to -39 | -77.4 | -87.2 to -67.3 | -3.8 | -7.1 to -0.9 |
| Longwoods | -28.8 | -34.4 to -23.1 | -15.2 | -22.9 to -6.5 | -97.4 | -110.2 to -87.4 | -221.5 | -240.2 to -202.1 | -27.6 | -32.7 to -23.3 |
| Egbert | -41.8 | -49.3 to -35.5 | -34.2 | -43.2 to -23.6 | -93.9 | -108.9 to -81.5 | -199.4 | -221.6 to -177.3 | -25.1 | -31.2 to -19.5 |
| Sprucedale | -32.8 | -52.4 to -12.6 | -29.5 | -39.9 to -16.6 | -99.0 | -150.5 to -36.8 | -140.0 | -188.7 to -97.2 | -54.4 | -74.5 to -36.3 |
| Chalk River | -8.8 | -11.6 to -6.4 | 2.8 | 1.5 to 4.3 | -59.8 | -66.9 to -52.1 | -104.7 | -115.3 to -94.1 | -8.2 | -10.8 to -5.7 |
| Chapais | -5.7 | -7 to -4.2 | 1.3 | 0.8 to 2 | -41.3 | -46.3 to -36 | -45.5 | -52.3 to -39.6 | -4.3 | -5.8 to -3.1 |
| Frelighsburg | -58.2 | -69 to -45.4 | -53.4 | -68.6 to -38.9 | -108.6 | -139.5 to -84 | -160.8 | -206 to -119.7 | -81.7 | -91.2 to -68.9 |
| Sutton | -4.6 | ns | 18.8 | 13.9 to 24.2 | -64.2 | -80.4 to -51.2 | -90.1 | -112.8 to -67.3 | -12.7 | -18.9 to -4 |
| Lac Edouard | -20.3 | -27.9 to -13.1 | -11.0 | -14.3 to -6.4 | -64.8 | -88 to -47.6 | -55.8 | -76.2 to -34.1 | -35.9 | -42.4 to -27.1 |
| Montmorency | 9.6 | 6.1 to 14.6 | 5.9 | 4.5 to 8.2 | -1.0 | ns | -20.1 | -40 to -3.2 | 21.0 | 14.9 to 29 |
| Kejimkujik | -5.3 | -6.5 to -4 | 2.5 | 1.4 to 3.8 | -53.1 | -59 to -47.5 | -35.9 | -41.2 to -30 | -6.5 | -7.9 to -4.8 |





Table 3: Rate of change in annual wet deposition of $Ca^{2+}$, $Mg^{2+}$, $K^+$, $Na^+$, and $Cl^-$. Slope refers to the Sen's slope (kg ha$^{-1}$ a$^{-1}$); C.I. refers to the 90% confidence interval of the slope; ns indicates no significant trend.

| Site | $Ca^{2+}$ | | $Mg^{2+}$ | | $K^+$ | | $Na^+$ | | $Cl^-$ | |
|---|---|---|---|---|---|---|---|---|---|---|
| | Slope | C.I. | Slope | C.I. | Slope | C.I. | Slope | C.I. | Slope | C.I. |
| Saturna | 0.001 | ns | -0.0003 | ns | -0.0002 | ns | 0.001 | ns | 0.002 | ns |
| Esther | 0.010 | ns | 0.003 | ns | 0.002 | ns | 0.002 | ns | -0.001 | ns |
| Cree Lake | -0.014 | ns | -0.001 | ns | -0.003 | ns | 0.001 | ns | -0.002 | ns |
| Bratt's Lake | 0.036 | ns | 0.009 | ns | 0.005 | ns | 0.002 | ns | 0.001 | ns |
| ELA | 0.008 | ns | 0.002 | ns | 0.002 | ns | -0.002 | -0.004 to -0.0002 | -0.001 | ns |
| Algoma | -0.009 | ns | -0.003 | -0.0064 to -0.0005 | -0.003 | -0.0059 to -0.0003 | -0.004 | ns | -0.015 | -0.025 to -0.01 |
| Longwoods | -0.017 | ns | -0.002 | ns | 0.006 | 0.003 to 0.011 | 0.004 | ns | -0.013 | -0.023 to -0.003 |
| Egbert | 0.003 | ns | -0.001 | ns | -0.0005 | ns | 0.004 | 0.002 to 0.009 | -0.008 | ns |
| Sprucedale | -0.024 | ns | -0.008 | -0.015 to -0.003 | -0.002 | ns | -0.005 | ns | -0.037 | ns |
| Chalk River | 0.001 | ns | -0.001 | ns | -0.001 | ns | 0.001 | ns | -0.008 | -0.013 to -0.004 |
| Chapais | 0.004 | ns | -0.001 | -0.0022 to -0.0003 | -0.0002 | ns | -0.003 | -0.006 to -0.001 | -0.007 | -0.012 to -0.002 |
| Frelighsburg | 0.020 | ns | 0.001 | ns | 0.001 | ns | 0.002 | ns | -0.012 | ns |
| Sutton | -0.029 | -0.058 to -0.005 | -0.002 | ns | -0.001 | ns | 0.002 | ns | -0.006 | ns |
| Lac Edouard | -0.004 | ns | -0.002 | ns | -0.002 | ns | -0.014 | -0.022 to -0.001 | -0.025 | -0.046 to -0.008 |
| Montmorency | -0.023 | ns | -0.002 | ns | 0.001 | ns | 0.009 | 0.0009 to 0.016 | 0.006 | ns |
| Kejimkujik | 0.007 | ns | 0.003 | ns | -0.0003 | ns | 0.024 | ns | 0.056 | ns |





Table 4: Rate of change in annual wet deposition of $NH_4^+$, $NO_3^-$, $SO_4^{2-}$, nss-$SO_4^{2-}$, precipitation amount and pH. Slope refers to the Sen's slope (kg ha$^{-1}$ a$^{-1}$); C.I. refers to the 90% confidence interval of the slope; ns indicates no significant trend; na indicates no available data.

| Site | $NH_4^+$ | | $NO_3^-$ | | $SO_4^{2-}$ | | nss-$SO_4^{2-}$ | | Annual precip (mm a$^{-1}$) | | pH (a$^{-1}$) | |
|---|---|---|---|---|---|---|---|---|---|---|---|---|
| | Slope | C.I. | Slope | C.I. | Slope | C.I. | Slope | C.I. | Slope | C.I. | Slope | C.I. |
| Saturna | -0.01 | ns | -0.07 | -0.13 to -0.01 | -0.13 | -0.17 to -0.06 | -0.12 | -0.16 to -0.07 | -1.5 | ns | 0.012 | 0.009 to 0.016 |
| Esther | 0.02 | ns | 0.04 | ns | -0.02 | ns | na | na | -1.4 | ns | -0.012 | ns |
| Cree Lake | -0.02 | ns | -0.05 | ns | -0.09 | ns | na | na | -2.0 | ns | -0.010 | ns |
| Bratt's Lake | 0.07 | ns | 0.02 | ns | 0.10 | ns | na | na | 25.9 | ns | -0.009 | ns |
| ELA | 0.05 | 0.03 to 0.07 | 0.02 | ns | -0.04 | ns | na | na | 6.0 | ns | 0.009 | 0.003 to 0.014 |
| Algoma | -0.06 | -0.09 to -0.02 | -0.38 | -0.51 to -0.26 | -0.60 | -0.71 to -0.47 | na | na | -7.8 | -14.5 to -2.7 | 0.020 | 0.016 to 0.025 |
| Longwoods | 0.03 | ns | -0.33 | -0.48 to -0.17 | -0.55 | -0.7 to -0.36 | na | na | 3.9 | ns | 0.023 | 0.019 to 0.028 |
| Egbert | -0.001 | ns | -0.31 | -0.45 to -0.18 | -0.45 | -0.57 to -0.33 | na | na | 3.8 | ns | 0.027 | 0.022 to 0.03 |
| Sprucedale | -0.03 | ns | -1.01 | -1.4 to -0.48 | -1.00 | -1.46 to -0.36 | na | na | 9.1 | ns | 0.030 | 0.018 to 0.042 |
| Chalk River | 0.01 | ns | -0.19 | -0.24 to -0.11 | -0.36 | -0.43 to -0.29 | na | na | 5.3 | 2.1 to 8.3 | 0.020 | 0.018 to 0.022 |
| Chapais | -0.01 | ns | -0.21 | -0.28 to -0.11 | -0.34 | -0.45 to -0.25 | na | na | -0.9 | ns | 0.015 | 0.013 to 0.018 |
| Frelighsburg | -0.02 | ns | -0.93 | -1.29 to -0.64 | -0.93 | -1.57 to -0.16 | na | na | 13.5 | ns | 0.056 | 0.036 to 0.069 |
| Sutton | 0.05 | 0.01 to 0.09 | 0.01 | ns | -0.57 | -0.9 to -0.23 | na | na | 4.0 | ns | 0.017 | 0.01 to 0.026 |
| Lac Edouard | -0.07 | ns | -0.69 | -0.91 to -0.38 | -0.46 | -0.77 to -0.23 | na | na | 14.6 | ns | 0.027 | 0.018 to 0.045 |
| Montmorency | 0.05 | ns | 0.20 | ns | -0.27 | ns | -0.27 | ns | 27.8 | 6.1 to 45.7 | 0.008 | 0.001 to 0.014 |
| Kejimkujik | 0.01 | ns | -0.12 | -0.18 to -0.06 | -0.27 | -0.37 to -0.2 | -0.28 | -0.36 to -0.2 | 10.9 | 3 to 17.2 | 0.014 | 0.011 to 0.017 |




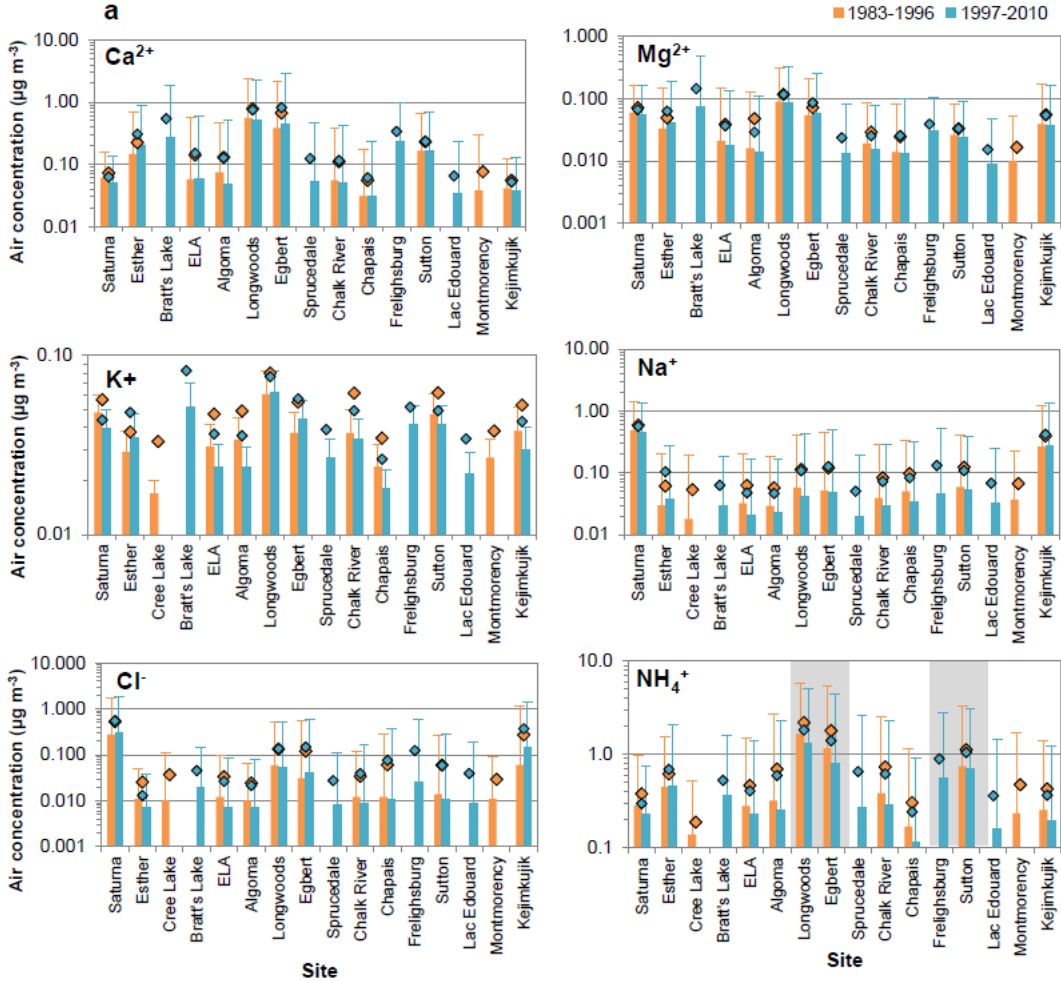





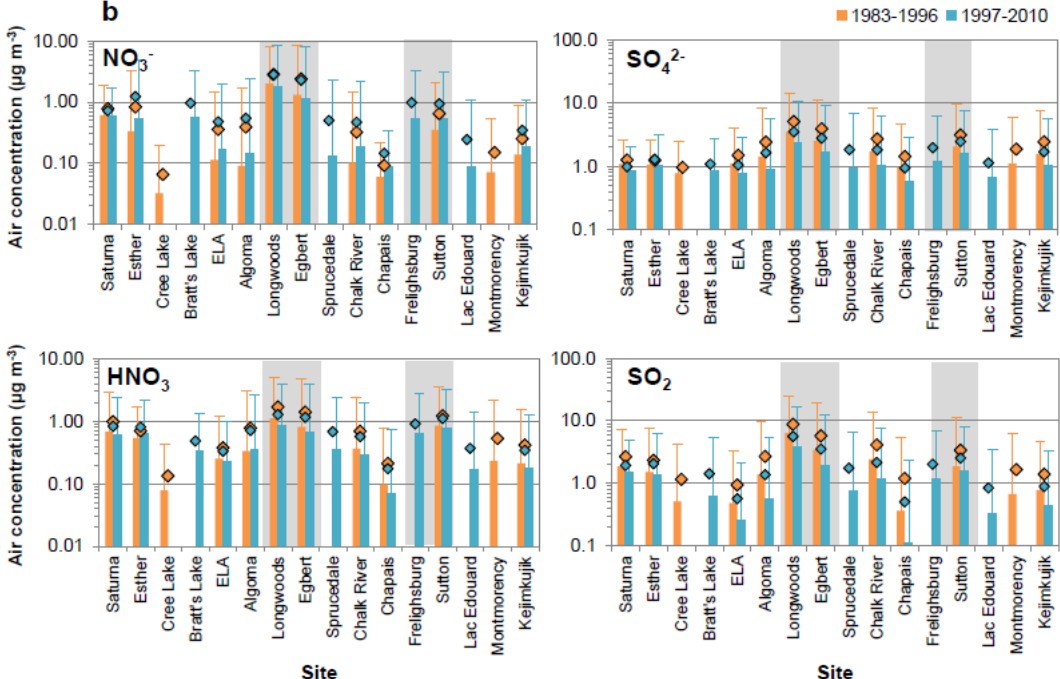

Figure 1: Geographical patterns of base cations, Cl⁻ and $NH_4^+$ (a) and $NO_3^-$, $SO_4^{2-}$, $HNO_3$ and $SO_2$ (b) for the 1983-1996 and 1997-2010 periods. Sites are arranged in order from western to eastern Canada. Bars indicate the median concentration. Error bars indicate the 95th percentile concentration. Diamonds indicate the mean concentration. Grey shaded regions correspond to southern Ontario or southern Quebec sites.



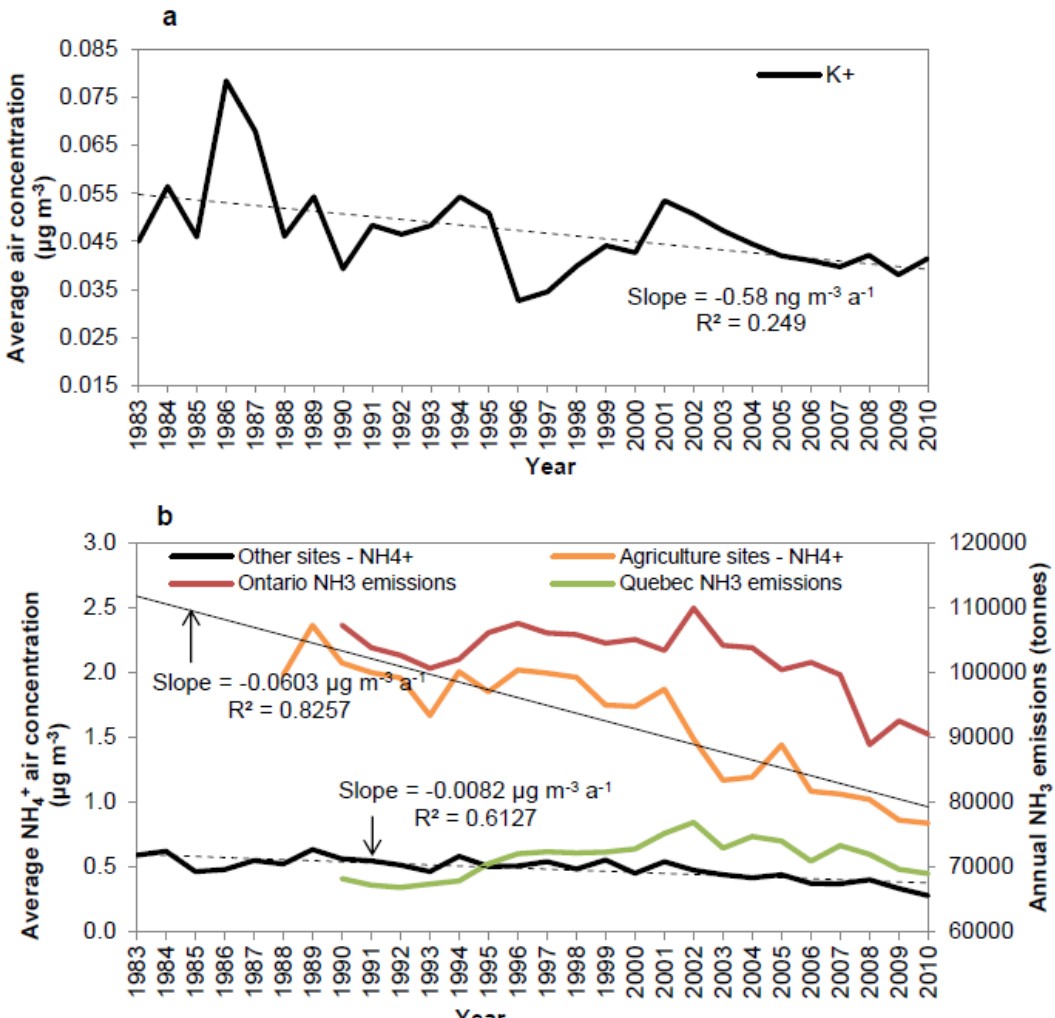

**Figure 2: Temporal trends of annual average atmospheric K⁺ (a) and atmospheric NH₄⁺ and annual ammonia emissions (b).**





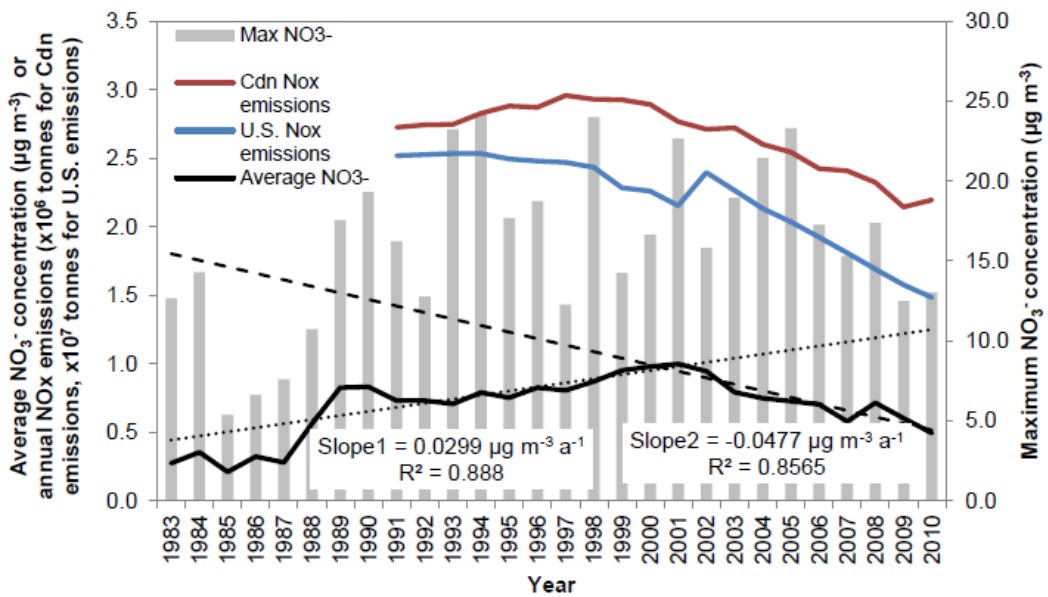

Figure 3: Temporal trends of annual atmospheric $NO_3^-$ and annual $NO_x$ emissions. Slope1 refers to the regression line for $NO_3^-$ between 1991 and 2001 (positive trend), while Slope2 is for the period between 2001 and 2010 (negative trend).

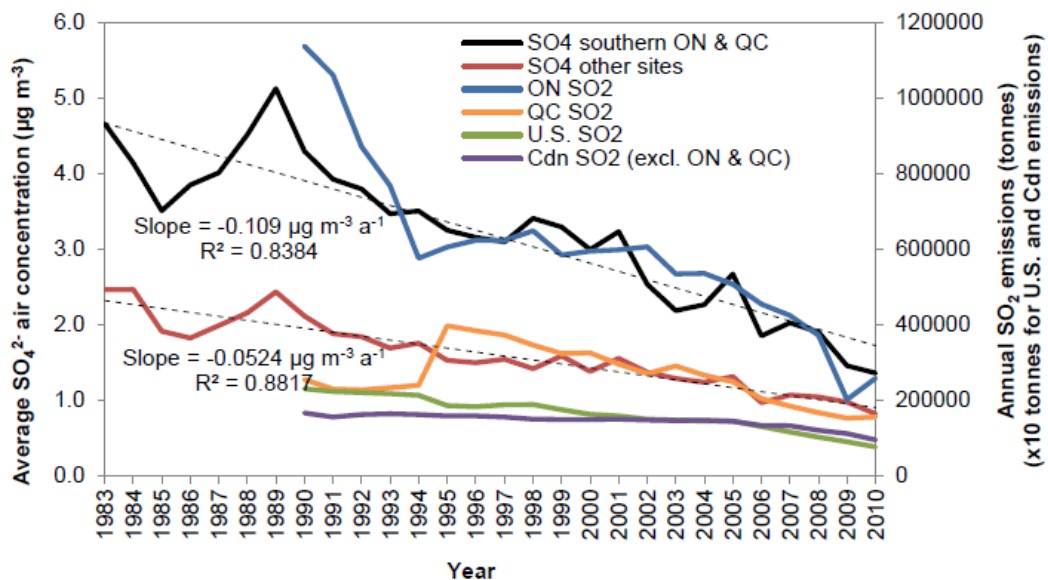

**Figure 4: Temporal trends of annual average atmospheric SO$_4^-$ and annual SO$_2$ emissions. ON and QC refer to the province of Ontario and Quebec, respectively.**





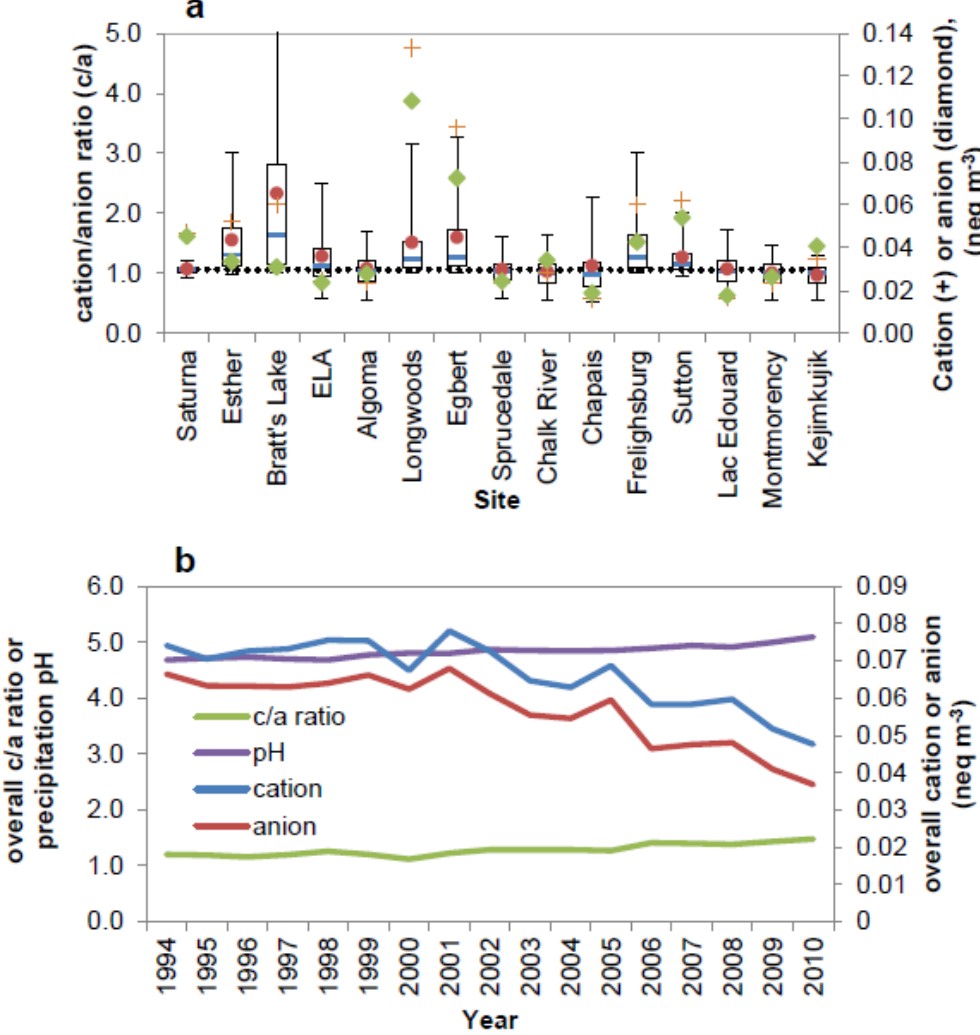

**Figure 5: Geographical patterns of cation/anion (c/a) ratio and cation and anion concentrations (a) and temporal trends of annual average c/a ratio, cation and anion concentrations, and precipitation pH (b). In (a), the blue line indicates the median; the red dot indicates the mean; the box and whiskers include the interquartile range and the 5th to 95th percentile range, respectively; the dotted line is the overall median among the sites.**







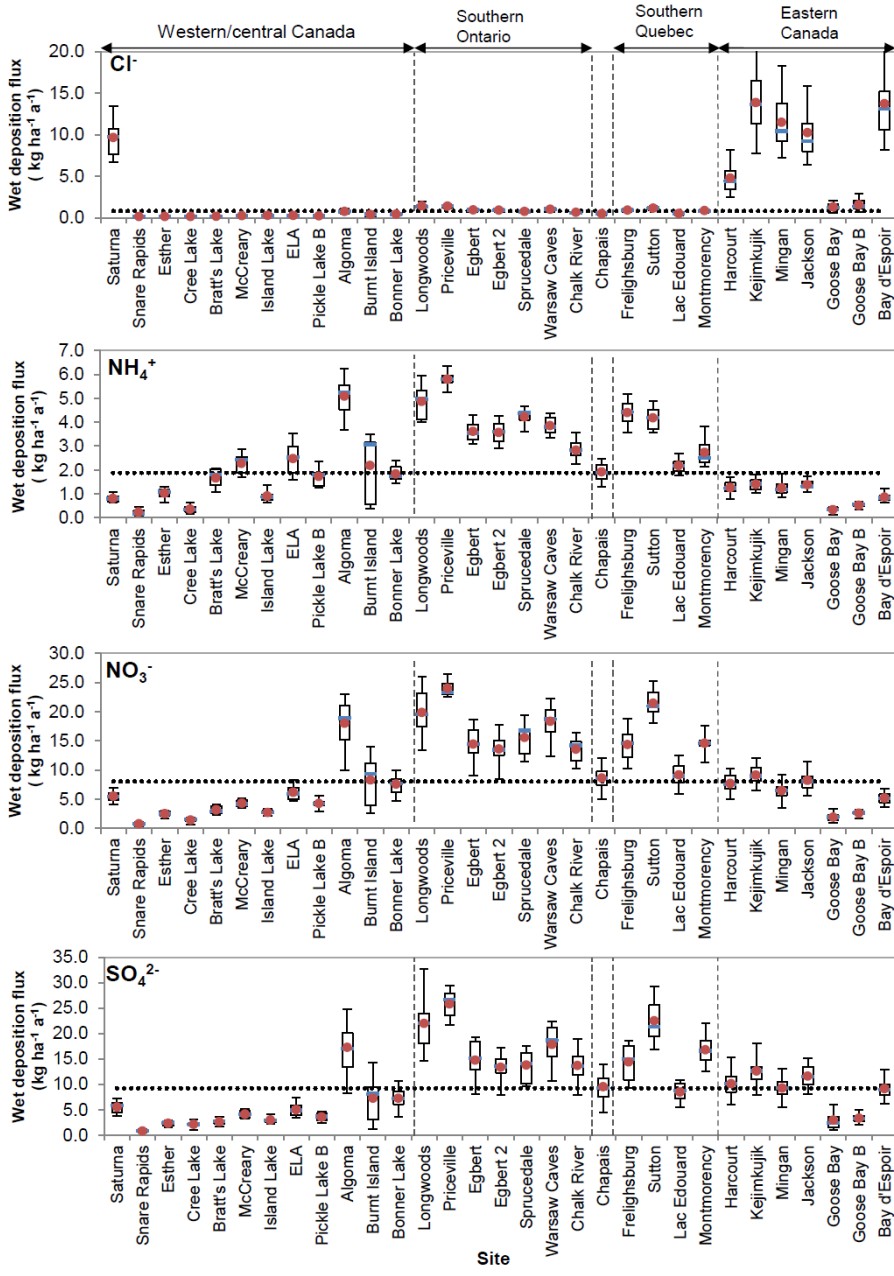

**Figure 6: Geographical patterns of the annual wet deposition of ions. Sites are arranged in order from western to eastern Canada. The blue line indicates the median; the red dot indicates the mean; the box and whiskers include the interquartile range and the 5th to 95th percentile range, respectively; the dotted line is the overall median among the sites.**



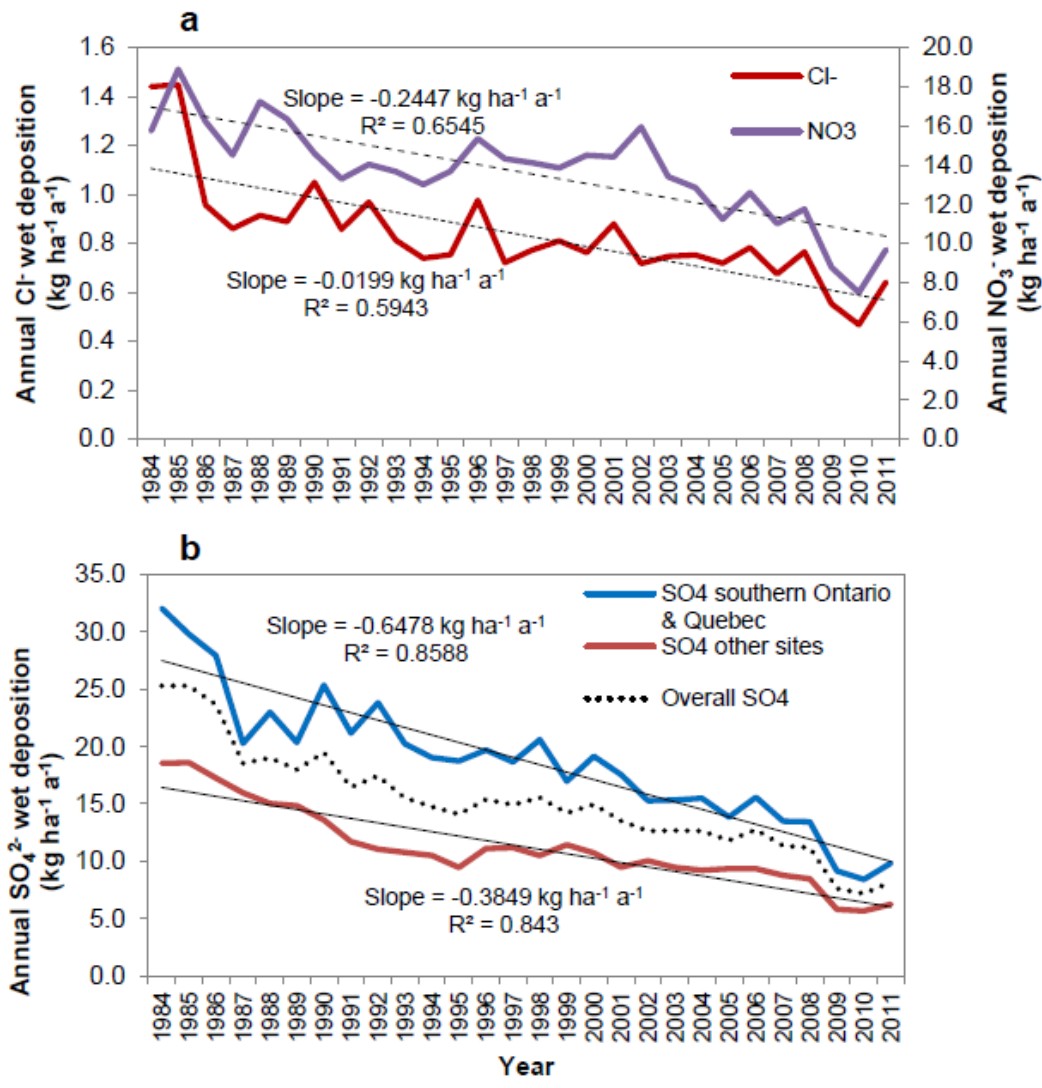

**Figure 7: Temporal trends of annual wet deposition of Cl⁻ and NO₃⁻ (a) and SO₄²⁻ (b).**





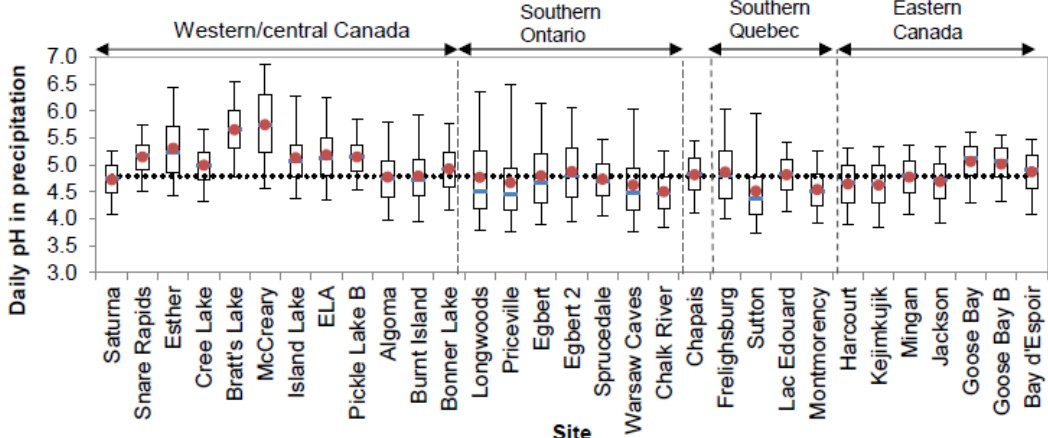

Figure 8: Geographical patterns of precipitation pH. See Fig. 6 caption.





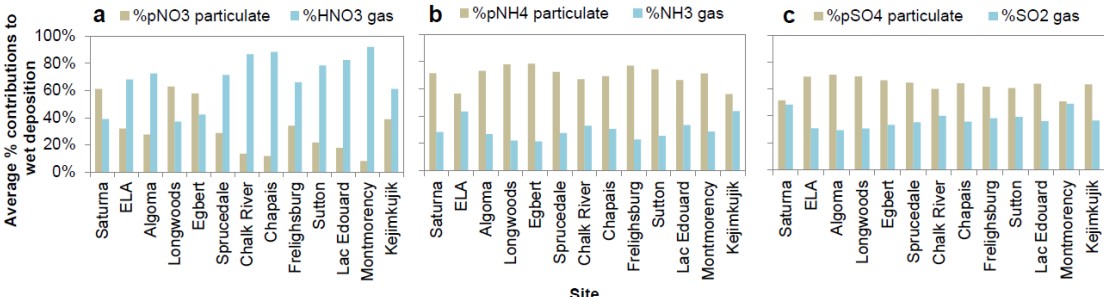

**Figure 9: Average percent contributions of gas and particulate-phase species to nitrate (a), ammonium (b), and sulfate (c) wet deposition.**





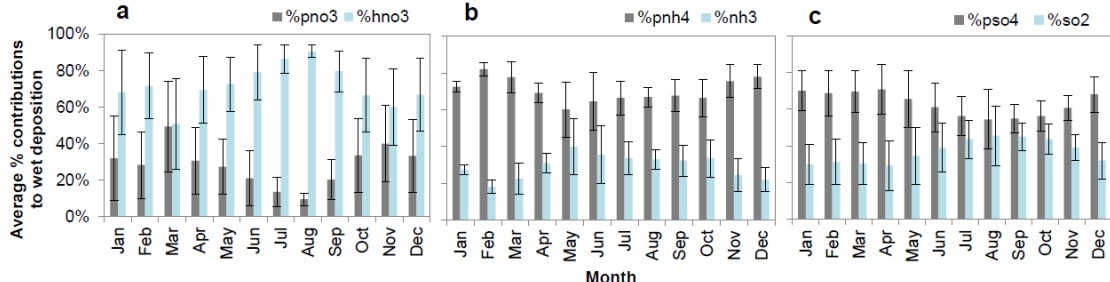

**Figure 10: Monthly trend in the contributions of gas and particulate-phase species to nitrate (a), ammonium (b), and sulfate (c) wet deposition. Error bars represent the standard deviation of the percent contributions by gas and particulate-phase species between sites.**