# Peer review of "Long-term air concentrations, wet deposition, and scavenging ratios of inorganic ions, HNO3 and SO2 and assessment of aerosol and precipitation acidity at Canadian rural locations"

_Atmospheric Chemistry and Physics, 2016_

## Referee Comment (RC1) · Anonymous Referee #1 · 26 Dec 2016

General comments The manuscript includes updated long-term data on concentrations, wet deposition, and scavenging ratios of atmospheric pollutants in Canadian sites. As outlined briefly in the introduction, there is, indeed, a need for keeping track of the recent evolution of atmospheric pollutants contributing to smog and acid rain, particularly in Canadian sites. Much of the information in the literature refers to datasets from the contiguous US or from European sites, two other regions where data from similar extensive monitoring networks have been available for decades, while Canadian sites have received relatively less attention. Observations of a recent decrease in precipitation acidity in North America (including Canada) and Europe have been accumulating, and have been the subject of many papers. The topic is not necessarily new. However, many aspects of the chemistry, transport, and deposition of these atmospheric pollutants are still not well-understood. Despite the inherent interest in reporting updated geographical patterns and temporal trends of atmospheric pollutants, much of the data presented here is not necessarily "new", and similar reports and conclusions can be found in the literature. For example, a recent global assessment of precipitation chemistry and deposition of these substances already includes much of the information presented in the manuscript (Vet et al. 2014). The report from Vet et al. (2014) not only includes much of the data used here (data obtained from the same CAPMoN network in the same locations), but also allows to put the data into a regional context, and compare the observed geographical patterns and temporal trends with those of emissions. Surprisingly, the Vet et al. (2014) assessment was not mentioned in the manuscript. In my opinion further attention should have been paid to clarify, complement and update the information already existing in the literature. I would recommend authors further efforts summarizing and presenting measurement data and trends in a clearer geographical/temporal context. According to the four objectives presented in the introduction some of the questions that needed to be answered in the manuscript were for example: as of 2011, how much have concentrations/deposition of the several substances decreased compared to a baseline year? Where? When? Are those changes parallel to emission reductions? Much of the text is a never-ending compilation of ranges, averages, medians and percentiles of the various substances, sometimes referred as measured in site X or site Y (a non-Canadian reader must check supplementary Figure S1 constantly to figure out these locations), without a clear message being told. The reader gets usually overwhelmed by the amount of values, ranges and percentages presented in the text, much of which could just have been summarized in figures and tables, while getting very little or disperse information about the magnitude of changes that have taken place, and the spatial and temporal context of

those changes. In my opinion, the major contribution of the manuscript is the calculation of scavenging ratios and the development of an approach to estimate particulate and gaseous wet scavenging contributions to wet deposition at those sites. Again, issues arise regarding the way scavenging data is presented and contextualized. Is there any clear spatial pattern in scavenging ratios? Have those ratios change as concentrations decreased over the decades? Have the relative contributions of particulate and gaseous substances to deposition changed over time? Specific comments Please avoid including too much data (if any) in the abstract. The message is lost among the many references to substances, ranges, years, . . . In section 2.1.4. authors stated that meteorological data was collected. Apart from precipitation, were these data included in the analyses? Are there any relationships between pollutants and wind, relative humidity or temperature data? Figure 1 is a very poor attempt to show the geographical and temporal changes in concentrations. First of all, the log scale makes it difficult to note the differences among sites or the changes between the two periods. Furthermore, what's the reason behind the two periods (1983-1996 and 1997-2010)? Is 1997 a "landmark" for Canadian emission regulations that would define a "before" and "after"? Why not comparing a year in the 1980's and a year at the end of the series (2011?) to actually show the decreases? In many cases (e.g. for NO3) the graphs show little changes or even increases, while in the text it has been clearly stated that concentrations of most substances have decreased. The case of NO3 is particularly interesting. Figure 3 shows that averaged NO3 concentrations and emissions started to decrease in 2001. Why not using that year to compare "before" and "after" concentrations for that substance? The multi-axis panels in Figure 5 are probably not the best option here. I would recommend to use cation/anion ratio data only for panel a (concentrations can be extracted from tables and text), and c/a ratio and pH data only for panel b (this time 2 Y axes can be used to highlight temporal trends). As mentioned in the general comments, some of the most interesting contributions of the manuscript are in my opinion those regarding scavenging ratios. Much of that information is included as supplementary materials. Some of those graphs and tables could have been part of

the main body text. References cited Vet, R., R. S. Artz, S. Carou, M. Shaw, C. U. Ro, W. Aas, A. Baker, V. C. Bowersox, F. Dentener, C. Galy-Lacaux, A. Hou, J. J. Pienaar, R. Gillett, M. C. Forti, S. Gromov, H. Hara, T. Khodzher, N. M. Mahowald, S. Nickovic, P. S. P. Rao, and N. W. Reid. 2014. A global assessment of precipitation chemistry and deposition of sulfur, nitrogen, sea salt, base cations, organic acids, acidity and pH, and phosphorus. Atmospheric Environment 93:3–100.

———————————————

---

## Referee Comment (RC2) · Anonymous Referee #2 · 26 Jan 2017

This manuscript presents a detailed analysis of the trace gas concentrations and wet deposition fluxes measured at the Canadian Air and Precipitation Monitoring Network (CAPMoN) sites in the past thirty years (1983-2011). Long-term trends of aerosol ions and precipitation acidity at the Canadian sites are assessed from the aspect of acid deposition. The scavenging ratios for air pollutants as a measure of the wet scavenging efficiency are also estimated and compared with previous studies. The manuscript also presented a first attempt to quantify the relative contribution of gas and particle to

measured wet deposition fluxes of ammonium, sulfate and nitrate.

Overall, the manuscript presents a very comprehensive summary of the air pollutant and wet deposition measurements at the Canadian network. While I also think that the manuscript can be difficult to follow due to the intense statistics reported in a parallel way, it is valuable to document the thirty-year dataset over Canada and the spatial and temporal patterns. The approach to estimate gas vs. aerosol contributions to wet deposition fluxes will be particularly helpful for model evaluation.

**Specific comments** 1) Abstract, Page 1, Line 19: Suggest change "because of the exclusion of gas scavenging" to "because of the exclusion of gas scavenging in previous studies" to avoid confusion.

2) Page 5, Line 10-13: This sentence is unclear. What is the difference between the Seasonal Kendall test and the Mann-Kendall test? Why are they inconsistent? Please clarify.

3) Page 5, Line 13-14 Page 10, last paragraph: "The relationship between meteorological factors and temporal trends in air concentrations and wet deposition were also examined by correlation analysis". As shown in Table S1 and discussed on Page 10, the correlation analysis was only applied to some particulate ions and trace gases, and not to wet deposition fluxes.
It is not convincing that we can use the correlation coefficients based on monthly averages to explain the long-term trends based on annual values. For example, the weak correlation between monthly K+ concentration and precipitation may largely differ from their annual correlation, which is a better indicator for the descending trends in K+ concentrations during 1993-2010. Please clarify.
Also please state in the title of Table S1 that the correlations are computed using monthly values.

4) Page 6, Line 7: Please clarify here how W(K/2) as the scavenging ratio for coastal site is calculated here. As half of W(K)?

5) Page 7, Equ. (4)(5): Replace the * with cross using the standard equation format.

6) Page 10, Figure 2-4: The manuscript concludes that the temporal trends in atmospheric inorganic concentrations are consistent with their emissions. Can you please also present the trends in percentage to discuss that to what extent the trends in pollutant concentrations reflect trends in emissions?

7) Page 34, label of Figure 3: Please explain 'Cdn' in the caption. And 'Nox' should be 'NOx'.

8) Page 42, caption of Figure 10: Suggest change here (and a few places in the text, e.g., seasonal trend) "monthly trend" to "monthly variation" to avoid confusion with the long-term trend.

9) Supplement, Page S5, caption of Figure S4: "Five different months" should be "five different years".

---

## Author Response (AR1)

Dear Prof. Yu (Editor):

We are submitting a revision of our paper (manuscript #acp-2016-918) entitled, "Long-term air concentrations, wet deposition, and scavenging ratios of inorganic ions, HNO3 and SO2 and assessment of aerosol and precipitation acidity at Canadian rural locations", for further consideration in Atmospheric Chemistry and Physics. We have addressed all of the comments provided by the reviewers. The details can be found in our enclosed responses to the reviewers' comments. For your convenience, a copy of the paper with track-changes is attached.

Thank you for taking care of the review process for this paper.

Sincerely,

Irene Cheng and Leiming Zhang

Air Quality Research Division Environment and Climate Change Canada

**Response to Reviewer 1**

We thank the reviewer for the suggestions to improve our paper. Our point-by-point responses to each comment are shown in blue.

**General comments**

The manuscript includes updated long-term data on concentrations, wet deposition, and scavenging ratios of atmospheric pollutants in Canadian sites. As outlined briefly in the introduction, there is, indeed, a need for keeping track of the recent evolution of atmospheric pollutants contributing to smog and acid rain, particularly in Canadian sites. Much of the information in the literature refers to datasets from the contiguous US or from European sites, two other regions where data from similar extensive monitoring networks have been available for decades, while Canadian sites have received relatively less attention. Observations of a recent decrease in precipitation acidity in North America (including Canada) and Europe have been accumulating, and have been the subject of many papers. The topic is not necessarily new. However, many aspects of the chemistry, transport, and deposition of these atmospheric pollutants are still not well-understood. Despite the inherent interest in reporting updated geographical patterns and temporal trends of atmospheric pollutants, much of the data presented here is not necessarily "new", and similar reports and conclusions can be found in the literature. For example, a recent global assessment of precipitation chemistry and deposition of these substances already includes much of the information presented in the manuscript (Vet et al. 2014). The report from Vet et al. (2014) not only includes much of the data used here (data obtained from the same CAPMoN network in the same locations), but also allows to put the data into a regional context, and compare the observed geographical patterns and temporal trends with those of emissions. Surprisingly, the Vet et al. (2014) assessment was not mentioned in the manuscript. In my opinion further attention should have been paid to clarify, complement and update the information already existing in the literature.

Response: The global assessment study by Vet et al. (2014) has been discussed in the revised paper (sect. 3.2.1 and 3.2.3). Please note that our study has a different scope from that of Vet et al.; thus, the data and discussion are presented in very different ways. Our study includes the analysis of geographical and temporal trends of not only wet deposition, but also the atmospheric concentrations (sect. 3.1). This difference is an important to consider because the atmospheric concentrations of SO2, sulfate, and nitrate have stronger links to emissions changes than wet deposition of sulfate and nitrate. The latter is primarily affected by particle sizes, air concentrations, rainfall intensity, and precipitation and cloud types, which varies geographically. The way the air concentration and wet deposition trends were compared to those of emissions were very different. Our study plotted the annual air concentrations and annual emissions on the same graph in parallel as shown in Figures 2b, 3 and 4, whereas the emissions from two points in time were analyzed by Vet et al. The second major difference is that our study provides more detailed analysis of trends specifically for Canadian sites and regions, whereas a much broader (global) context of the trends were presented by Vet et al. over two three-year time periods (2000-2002 and 2005-2007). In this study, we compared the trends obtained from our study to those already existing in literature, e.g. Zbieranowski and Aherne (2011) for nitrogen measurements at Canadian sites and Lehmann et al. (2005, 2007) for U.S. sites. These two

studies were discussed in more detail with our results because we applied the same statistical method to analyze the long term temporal trends (Seasonal Kendall test and Mann-Kendall test).

I would recommend authors further efforts summarizing and presenting measurement data and trends in a clearer geographical/temporal context. According to the four objectives presented in the introduction some of the questions that needed to be answered in the manuscript were for example: as of 2011, how much have concentrations/deposition of the several substances decreased compared to a baseline year? Where? When? Are those changes parallel to emission reductions?

Response: We summarized the geographical and temporal trends as much as possible despite the vast amounts of data from the CAPMoN sites. Regarding geographical patterns, we reported the range in the median air concentration and range in annual wet deposition and then discussed reasons for the spatial variation (e.g. differences between coastal and continental sites, agricultural sites, and sites impacted by anthropogenic emissions). In terms of temporal trends, we reported the direction of the long-term trends found at most of the sites (increasing/decreasing) and the range in the magnitude of the trends for air concentrations and annual wet deposition. We then discussed the possible reasons (e.g. NH3, SO2, NOx emissions reductions) for particularly larger declines at southern Ontario and southern Quebec sites for atmospheric ammonium, sulfate and nitrate. This discussion was supported by showing a graph of the annual trends of air concentrations and emissions that are shown in parallel (Figures 2b, 3 and 4). The method for determining trends was not a comparison of the 2011 data with a baseline year. In our opinion, this method only gives a rough estimate of the change in the air concentrations or wet deposition between two points in time because it does not take into account the year-to-year variability or the statistical significance of the trends. Thus, in this study we examined the long-term trends using a well-established method for conducting statistical trends analysis (Seasonal Kendall test and Mann-Kendall test).

Much of the text is a never-ending compilation of ranges, averages, medians and percentiles of the various substances, sometimes referred as measured in site X or site Y (a non-Canadian reader must check supplementary Figure S1 constantly to figure out these locations), without a clear message being told. The reader gets usually overwhelmed by the amount of values, ranges and percentages presented in the text, much of which could just have been summarized in figures and tables, while getting very little or disperse information about the magnitude of changes that have taken place, and the spatial and temporal context of those changes.

Response: In terms of quantitative information, the text reported the range in the median air concentration and the range in annual wet deposition among the sites for the discussion on geographical variability. For temporal variability, the text reported only the range in the magnitude of the trends for air concentrations and annual wet deposition among the sites. Similarly only the range was reported for aerosol acidity, acid rain and the scavenging ratios. Considering the large amount of data analyzed, we feel that the statistics have already been minimized in the text. The detailed statistics for each site are provided in the tables and figures.

The text provided context on the spatial and temporal patterns and did not only refer to specific sites. For geographical patterns, we discussed that the spatial variation were attributed to sea salt

emissions from the ocean, agricultural activities, and anthropogenic emissions. We also mentioned the names of the sites belonging to these categories in case readers are interested in the detailed statistics for these sites in the tables and figures. For temporal trends, we discussed that the stronger declining trends at southern Ontario and southern Quebec sites for atmospheric ammonium, sulfate and nitrate were attributed to NH3, SO2, NOx emissions reductions. We showed aerosol acidity was important to eastern Canada and central Ontario than other Canadian regions. We also discussed that acid rain impacts were more significant in southern Ontario and eastern Canada than western and central Canada. These examples show that the spatial and temporal trends were discussed in a broader, regional context in the paper, and not only referring to the individual sites.

In my opinion, the major contribution of the manuscript is the calculation of scavenging ratios and the development of an approach to estimate particulate and gaseous wet scavenging contributions to wet deposition at those sites. Again, issues arise regarding the way scavenging data is presented and contextualized. Is there any clear spatial pattern in scavenging ratios? Have those ratios change as concentrations decreased over the decades? Have the relative contributions of particulate and gaseous substances to deposition changed over time?

Response: The spatial patterns in scavenging ratios are discussed in the first paragraph of sect. 3.3.2. In this paragraph, we attributed the spatial variability in scavenging ratios to differences in the particle size distributions between coastal and inland sites.

We analyzed the long-term trends in the monthly scavenging ratios of  $pSO_4^{2-}$  and  $SO_2$  and  $pNO_3^{--}$ and HNO3 and their relative contributions to total wet deposition at eight sites representative of western/central Canada, southern Ontario and southern Quebec, and eastern Canada. Most of the trends were not statistically significant according to the Seasonal Kendall test. No consistent trends were found within each Canadian region either. Some statistically significant trends in scavenging ratios were found at a few locations and for some nitrogen and sulfur species. At Longwoods, there was a statistically significant declining trend in the scavenging ratio of  $pNO_3$ ; however the magnitude of the trend was only -6.3 (<1%) per year which is small compared to the scavenging ratio (values in the hundreds to thousands). At Algoma, a significant increasing trend in the scavenging ratio of  $pSO_4^{2-}$  was found with a slope of +11.4 per year. This suggests the atmospheric  $pSO_4^{2}$  concentrations were decreasing at a slightly faster rate than the decrease in precipitation sulfate. This result is consistent with the emissions reductions having a greater impact on air concentrations than precipitation concentrations. Although the trend is statistically significant, the magnitude of the trend (11.4 or 1.4% per year) is again very small. At many of the sites, the lack of long-term trends in the scavenging ratios of sulfur and nitrogen species reflect the decreasing trends in both wet deposition and air concentrations (Table 2 and 4). There are also many factors that can affect the precipitation concentrations, such as particle sizes, air concentrations, rainfall intensity, and precipitation and cloud types, which vary geographically and could change over time. These uncertainties make it difficult to narrow down the reasons behind the long-term scavenging ratio trends.

We added the following paragraph in sect. 3.3.2 of the revised paper: "Most of the long-term scavenging ratio trends were not statistically significant according to the Seasonal Kendall test, but some statistically significant trends were found at a few locations and for some nitrogen and

sulfur species. At Longwoods, there was a statistically significant declining trend in the scavenging ratio of  $pNO_3^-$ ; however the magnitude of the trend was only -6.3 (<1%) per year which is small compared to the scavenging ratio (values in the hundreds to thousands). At Algoma, a significant increasing trend in the scavenging ratio of  $pSO_4^{2-}$  was found with a slope of +11.4 (1.4%) per year. At many of the sites, the lack of long-term trends in the scavenging ratios of sulfur and nitrogen species reflect the decreasing trends in both wet deposition and air concentrations (Table 2 and 4). There are also many factors that can affect the precipitation concentrations, such as particle sizes, air concentrations, rainfall intensity, and precipitation and cloud types, which vary geographically and could change over time."

**Specific comments**

Please avoid including too much data (if any) in the abstract. The message is lost among the many references to substances, ranges, years

Response: The abstract has been revised by making it less quantitative. It now reads, "This study analyzed long-term air concentrations and annual wet deposition of inorganic ions and aerosol and precipitation acidity at 30 Canadian sites from 1983-2011. Scavenging ratios of inorganic ions and relative contributions of particulate- and gas-phase species to NH4+, NO3-, and SO42- wet deposition were determined. Geographical patterns of atmospheric  $Ca^{2+}$ ,  $Na^+$ ,  $Cl^-$ ,  $NH_4^+$ ,  $NO_3^{-1}$ , and  $SO_4^{-2-}$  were similar to wet deposition and attributed to anthropogenic sources, sea-salt emissions, and agricultural emissions. Decreasing trends in atmospheric  $NH_4^+$  (1994-2010) and  $SO_4^{2-}$  (1983-2010) were prevalent. Atmospheric NO3 increased prior to 2001 and then declined afterwards. These results are consistent with SO2, NOx and NH3 emission trends in Canada and the U.S. Widespread declines in annual NO3- and SO42- wet deposition ranged from 0.07-1.0 kg ha-1 a-1 (1984-2011). Acidic aerosols and precipitation impacted southern and eastern Canada more than western Canada; however both trends have been decreasing since 1994. Scavenging ratios of particulate  $NH_4^+$ ,  $SO_4^{2-}$  and  $NO_3^-$  differed from literature values by 22%, 44% and a factor of 6, respectively, because of the exclusion of gas scavenging in previous studies. Average gas and particle scavenging contributions to total wet deposition were estimated to be 72% for HNO3 and 28% for particulate NO3-, 37% for SO2 and 63% for particulate SO42-, and 30% for NH3 and 70% for particulate NH4+."

In section 2.1.4. authors stated that meteorological data was collected. Apart from precipitation, were these data included in the analyses? Are there any relationships between pollutants and wind, relative humidity or temperature data?

Response: Analysis of the relationships between the pollutants and temperature, relative humidity, and precipitation were discussed in the last paragraph of sect. 3.1.2 (p.10 ACPD version) and the correlation analysis results were shown in Table S1 of the supplementary material. We also showed the time-series trend of atmospheric  $K^+$  was similar to that of temperature for several agriculture/forested sites (Fig. S2a). In Fig. S2b, we showed the time-series trend of atmospheric ammonium was similar to that of temperature, and suggested that this was related to the formation of sulfate from the oxidation of sulfur dioxide based on the strong correlation between the sulfate/sulfur dioxide ratio and temperature. Relative humidity and precipitation had almost no influence on the temporal trends of particulate inorganic ions. For

wind direction data, we used it to confirm that the lower latitude sites (Longwoods, Egbert, Frelighsburg/Sutton) were impacted by transboundary emissions of  $SO_2$  and  $NO_x$  from the U.S. (p.8, lines 17-22 ACPD version). The wind direction data was also used to estimate the percentage frequency of winds from the ocean, which are contributing to atmospheric Na+ and Cl- at coastal sites (Saturna, Kejimkujik) (p.8, lines 6-11 ACPD version).

Figure 1 is a very poor attempt to show the geographical and temporal changes in concentrations. First of all, the log scale makes it difficult to note the differences among sites or the changes between the two periods. Furthermore, what's the reason behind the two periods (1983-1996 and 1997-2010)? Is 1997 a "landmark" for Canadian emission regulations that would define a "before" and "after"? Why not comparing a year in the 1980's and a year at the end of the series (2011?) to actually show the decreases?

Response: Figure 1 has been revised; the log scale has been changed to a standard scale and box plots (similar to Fig. 6 for annual wet deposition) are used to show more statistics. We also created separate graphs for the 1983-1996 and 1997-2010 periods to show the differences in the geographical distributions of the air concentration. The air concentrations from 1983-2010 were analyzed in this study; 1997 is the halfway point in the data. We chose to divide the data at the halfway point to ensure consistency in the sample size when comparing data between two time periods. The emission changes in Canada and the U.S. did not occur in one particular year; thus 1997 should not be viewed as a landmark year for emission changes. As discussed on p. 9-10 ACPD version, ammonia emissions in Ontario and Quebec decreased only after 2002. NOx emissions in Canada began to decrease in 1997, while in the U.S. the decline was already occurring in 1994. From Fig. 4, SO2 emissions in the U.S. have been declining since the early 1990s; however, in Canada only a larger decrease was observed after 2005. Given the vast amount of data analyzed in this study, there are numerous ways to analyze and present the data. Another way as the reviewer suggests is to compare the data from one year in the 1980s to another year in 2011. In our opinion, comparing one year to another year does not give very representative results. There may be year-to-year variability which is not considered when analyzing only two points in time. There also needs to strong justification on why the comparison was done specifically between year A and year B instead of other years.

In many cases (e.g. for NO3) the graphs show little changes or even increases, while in the text it has been clearly stated that concentrations of most substances have decreased. The case of NO3 is particularly interesting. Figure 3 shows that averaged NO3 concentrations and emissions started to decrease in 2001. Why not using that year to compare "before" and "after" concentrations for that substance?

Response: In the text, it states there was an even split in the number of sites with increasing and decreasing trends in  $NO_3^-$  (p. 9, lines 30 onwards, ACPD version). We also reported that at 9 of 16 sites an increasing trend was found between 1991 and 2001 which was followed by a decreasing trend from 2001 to 2010 (p.10, lines 2-3). The increasing and decreasing trends in  $NO_3^-$  were reflected in the Canadian  $NO_x$  emissions data, which showed an increased from 1991 to 1997 and then decreased from 1997 to 2010 (Fig. 3). This shows the consistency between the graphs (Fig. 3) and the discussion in the text. In Fig. 3, we plotted the annual trends in  $NO_3^-$  and

in  $NO_x$  emissions from 1991 to 2010, which clearly shows the change in the direction of the trends.

The multi-axis panels in Figure 5 are probably not the best option here. I would recommend to use cation/anion ratio data only for panel a (concentrations can be extracted from tables and text), and c/a ratio and pH data only for panel b (this time 2 Y axes can be used to highlight temporal trends).

Response: Figure 5 has been revised based on your suggestions. In the revised paper, Fig. 5a shows only the cation/anion (c/a) ratio, while Fig. 5b shows the c/a ratio and precipitation pH data. The cation and anion equivalent concentrations that were previously plotted in these figures have been summarized in Table S2.

As mentioned in the general comments, some of the most interesting contributions of the manuscript are in my opinion those regarding scavenging ratios. Much of that information is included as supplementary materials. Some of those graphs and tables could have been part of the main body text.

Response: There are already four large tables and ten figures in the main manuscript; thus there is not enough space to accommodate the tables of scavenging ratio statistics (in Table S2 and S3 of the supplementary material) or the large figure showing the monthly scavenging ratio variation (Fig. S5). Due to the different topics covered in this paper (including the geographical and temporal patterns of air concentrations, wet deposition, aerosol acidity, acid rain, scavenging ratios, and gas vs. particle wet scavenging), it is impossible to show all the results in the main manuscript.

**Response to Reviewer 2**

We thank the reviewer for the suggestions to improve our paper. Our point-by-point responses to each comment are shown in blue.

This manuscript presents a detailed analysis of the trace gas concentrations and wet deposition fluxes measured at the Canadian Air and Precipitation Monitoring Network (CAPMoN) sites in the past thirty years (1983-2011). Long-term trends of aerosol ions and precipitation acidity at the Canadian sites are assessed from the aspect of acid deposition. The scavenging ratios for air pollutants as a measure of the wet scavenging efficiency are also estimated and compared with previous studies. The manuscript also presented a first attempt to quantify the relative contribution of gas and particle to measured wet deposition fluxes of ammonium, sulfate and nitrate.

Overall, the manuscript presents a very comprehensive summary of the air pollutant and wet deposition measurements at the Canadian network. While I also think that the manuscript can be difficult to follow due to the intense statistics reported in a parallel way, it is valuable to document the thirty-year dataset over Canada and the spatial and temporal patterns. The approach to estimate gas vs. aerosol contributions to wet deposition fluxes will be particularly helpful for model evaluation.

**Specific comments**

1) Abstract, Page 1, Line 19: Suggest change "because of the exclusion of gas scavenging" to "because of the exclusion of gas scavenging in previous studies" to avoid confusion.

**Response: Revised according to your suggestion.**

2) Page 5, Line 10-13: This sentence is unclear. What is the difference between the Seasonal Kendall test and the Mann-Kendall test? Why are they inconsistent? Please clarify.

Response: The Seasonal Kendall test analyzes the temporal trend in the average air concentrations in each month separately and then aggregates the results to obtain the annual trend. In the Mann-Kendall test, the data are not split into twelve months before the temporal trend is analyzed (Gilbert, 1987). Therefore, the latter was used to obtain the annual total wet deposition trend. This has been clarified in sect. 2.2.1 of the revised paper.

3) Page 5, Line 13-14 Page 10, last paragraph: "The relationship between meteorological factors and temporal trends in air concentrations and wet deposition were also examined by correlation analysis". As shown in Table S1 and discussed on Page 10, the correlation analysis was only applied to some particulate ions and trace gases, and not to wet deposition fluxes. It is not convincing that we can use the correlation coefficients based on monthly averages to explain the long-term trends based on annual values. For example, the weak correlation between monthly K+ concentration and precipitation may largely differ from their annual correlation, which is a better indicator for the descending trends in K+ concentrations during 1993-2010.

Please clarify. Also please state in the title of Table S1 that the correlations are computed using monthly values.

Response: The air concentrations are more likely to be influenced by meteorological parameters (e.g. temperature, relative humidity, and precipitation) than wet deposition of pollutants. Wet deposition of the pollutants is predominantly affected by the air concentrations and the precipitation amount. Thus for wet deposition, we already discussed these correlation analysis results on p. 14 lines 16-32 of the ACPD version. To clarify, we revised the sentence before sect. 2.2.2 to, "Correlation analysis was performed between monthly averaged meteorological parameters and particulate ions and trace gases. For wet deposition of inorganic ions, the correlations with the precipitation amount and air concentrations were examined." The caption for Table S1 was also revised to, "Pearson correlation coefficients between selected atmospheric ions and meteorological parameters (significant at p<0.05; otherwise non-significant (ns)). Note that the ion concentrations and meteorological parameters are monthly averages."

Regarding the analysis of annual atmospheric  $K^+$  and annual precipitation amount, there was a statistically significant decreasing trend in atmospheric  $K^+$  for almost all the sites (Table 2, see the slope and C.I.); however, no statistically significant trends in the precipitation amount were found at the majority of the sites (Table 4, see the slope and C.I.). Note that it is important to consider the C.I. (90% confidence interval of the slope) to assess whether the trend is statistically significant. The results indicate the annual atmospheric  $K^+$  and annual precipitation trends were not the same.

4) Page 6, Line 7: Please clarify here how W(K/2) as the scavenging ratio for coastal site is calculated here. As half of W(K)?

Response: Atmospheric  $K^+$  has a bimodal particle size distribution. It is predominantly associated with fine particles at inland locations, but also associated with coarse sea-salt aerosols at coastal locations. In our previous study (Cheng et al., 2015), we observed that the mean scavenging ratio of fine particles was 34 to 52% of that of coarse particles at inland locations, but was 80% at the coastal sites. Therefore, the fine scavenging ratio was reduced by about a factor of 2 at coastal locations to take into account that the K+ that may be associated with coarse aerosols. This explanation has been added to sect. 2.2.3 of the revised paper.

5) Page 7, Equ. (4)(5): Replace the \* with cross using the standard equation format.

Response: Equations 4 and 5 have been revised according to your suggestions.

6) Page 10, Figure 2-4: The manuscript concludes that the temporal trends in atmospheric inorganic concentrations are consistent with their emissions. Can you please also present the trends in percentage to discuss that to what extent the trends in pollutant concentrations reflect trends in emissions?

Response: The temporal trends expressed as percentages have been added to the captions of Fig. 2 to 4. In Fig. 2b between 2002 and 2010 (as discussed in the text), atmospheric ammonium at agricultural sites decreased by 6.3% while ammonia emissions in Ontario and Quebec decreased

by 2.4% and 1.1%, respectively. For atmospheric nitrate and NOx emissions (Fig. 3), the timing of the trends were not synchronized as discussed in the text. In Fig. 3 between 1991 and 2001 (as discussed in the text), atmospheric nitrate increased by 3.6%. However, the period of increasing NOx emissions in Canada was between 1991 and 1997 (at 8.5%). Then, from 2001 to 2010, atmospheric nitrate decreased by 6.5%. However, the period of decreasing NOx emissions in Canada was between 1997 and 2010 (at 25.8%). In Fig. 4 between 1990 and 2010 (as discussed in the text), atmospheric sulfate decreased by 4.5% at southern Ontario and Quebec sites and by 3.6% at other sites. Over the same period, SO2 emissions decreased by 1.8% in Canada and 4.3% in the U.S. Note that the percentage change in pollutant concentrations may not be the same as the percentage change in emissions because the quantities are very different. However, it is clear that the direction of the trends is consistent.

7) Page 34, label of Figure 3: Please explain 'Cdn' in the caption. And 'Nox' should be 'NOx'.

Response: Cdn is the abbreviation used for Canada. We have added this explanation in the captions for Fig. 3 and 4 and changed Nox to NOx in the legend.

8) Page 42, caption of Figure 10: Suggest change here (and a few places in the text, e.g., seasonal trend) "monthly trend" to "monthly variation" to avoid confusion with the long-term trend.

Response: In the revised paper, supplementary material and Fig. 10, we replaced "seasonal trend" or "monthly trend" with "seasonal variation" or "monthly variation".

9) Supplement, Page S5, caption of Figure S4: "Five different months" should be "five different years".

Response: This has been corrected in the revised paper.

**Long-term air concentrations, wet deposition, and scavenging ratios of inorganic ions, HNO3 and SO2 and assessment of aerosol and precipitation acidity at Canadian rural locations**

Irene Cheng and Leiming Zhang

Correspondence to: Irene Cheng (irene.cheng@canada.ca)

**Abstract.** This study analyzed long-term air concentrations and annual wet deposition of inorganic ions and aerosol and precipitation acidity at 30 Canadian sites from 1983-2011. Scavenging ratios of inorganic ions and relative contributions of

- 10 particulate- and gas-phase species to  $NH_4^+$ ,  $NO_3^-$ , and  $SO_4^{2-}$  wet deposition were determined. Long term median atmospheric  $NH_4^+$ ,  $NO_3^-$ , and  $SO_4^{2-}$  between sites ranged from 0.1–1.7, 0.03–2.0, and 0.6–3.5 µg m-3, respectively. Their median annual wet deposition varied from 0.2–5.8, 0.8–23.3, and 0.8–26.6 kg ha-1 a-1. Geographical patterns of atmospheric Ca2+, Na+, Cl-,  $NH_4^+$ ,  $NO_3^-$ , and  $SO_4^{2-}$  were similar to wet deposition and attributed to anthropogenic sources, sea-salt emissions, and agricultural emissions. Decreasing trends in atmospheric  $NH_4^+$  (1994-2010) and  $SO_4^{2-}$  (1983-2010) were prevalent.
- Atmospheric NO3- increased from 1991-2001 prior to 2001 and then declined from 2001-2010 afterwards. These results are consistent with SO2, NOx and NH3 
[revised manuscript text omitted]

form at lower temperatures (Zhang et al., 2008; Zhao and Gao, 2008). A  $P_f$  of 0.84 was assumed for the winter months (DJF), whereas 0.29 was used for all other months. These are average mass fractions observed at CAPMoN sites in a short-term field study (Zhang et al., 2008). Eq. 2 accounts for the different scavenging efficiencies of small and large particles.

The contribution of HNO3 to nitrate wet deposition was calculated using Eq. 3:

5
$$[HNO_3]_{\text{prec}} = [total NO_3]_{\text{prec}} - [pNO_3]_{\text{prec}},$$
(3)

[total NO3-]prec is the monthly volume-weighted NO3-precipitation concentration and [pNO3-]prec is the wet scavenging of pNO3- calculated from Eq. 2. If [HNO3]prec< 0, it is assumed that only pNO3- contributed to total nitrate precipitation and no gas scavenging occurred. The relative contributions of particulate and gaseous species to NO3- wet deposition were determined using Eq. 4 and 5. Scavenging ratios of pNO3- and HNO3 were determined using Eq. 1.

10
$$\text{%pNO}_3^- = ([pNO_3^-]_{prec} / [total NO_3^-]_{prec})^* \underline{x}_100\%,$$
 (4)

$$% HNO_3 = ([HNO_3]_{\text{prec}}/[\text{total NO}_3^-]_{\text{prec}})^{\underline{*}} \underline{x} 100\%,$$
(5)

[revised manuscript text omitted]

| Site         | Ca 2+ |            | Mg 2+ |             | $\mathbf{K}^+$ |            | Na + |             | Cl     |                     |
|--------------|------------------|------------|------------------|-------------|----------------|------------|-----------------|-------------|--------|---------------------|
|              | Slope            | C.I.       | Slope            | C.I.        | Slope          | C.I.       | Slope           | C.I.        | Slope  | C.I.                |
| Saturna      | 0.001            | ns         | -0.0003          | ns          | -0.0002        | ns         | 0.001           | ns          | 0.002  | ns                  |
| Esther       | 0.010            | ns         | 0.003            | ns          | 0.002          | ns         | 0.002           | ns          | -0.001 | ns                  |
| Cree Lake    | -0.014           | ns         | -0.001           | ns          | -0.003         | ns         | 0.001           | ns          | -0.002 | ns                  |
| Bratt's Lake | 0.036            | ns         | 0.009            | ns          | 0.005          | ns         | 0.002           | ns          | 0.001  | ns                  |
| ELA          | 0.008            | ns         | 0.002            | ns          | 0.002          | ns         | -0.002          | -0.004 to - | -0.001 | ns                  |
|              |                  |            |                  |             |                |            |                 | 0.0002      |        |                     |
| Algoma       | -0.009           | ns         | -0.003           | -0.0064 to  | -0.003         | -0.0059 to | -0.004          | ns          | -0.015 | -0.025 to           |
| C            |                  |            |                  | -0.0005     |                | -0.0003    |                 |             |        | -0.01               |
| Longwoods    | -0.017           | ns         | -0.002           | ns          | 0.006          | 0.003 to   | 0.004           | ns          | -0.013 | -0.023 to           |
|              |                  |            |                  |             |                | 0.011      |                 |             |        | -0.003              |
| Egbert       | 0.003            | ns         | -0.001           | ns          | -0.0005        | ns         | 0.004           | 0.002 to    | -0.008 | ns                  |
| a 11         | 0.004            |            | 0.000            | 0.01 5      | 0.000          |            | 0.00 -   | 0.009       | 0.005  |                     |
| Sprucedale   | -0.024           | ns         | -0.008           | -0.015 to - | -0.002         | ns         | -0.005          | ns          | -0.037 | ns                  |
| Chalk Divor  | 0.001            | n c | 0.001            | 0.003       | 0.001          | ns         | 0.001           | ne          | 0.008  | 0.013 to            |
|              | 0.001            | 115        | -0.001           | 115         | -0.001         | 115        | 0.001           | 115         | -0.008 | -0.013 10           |
| Chanais      | 0.004            | ns         | -0.001           | -0.0022 to  | -0.0002        | ns         | -0.003          | -0.006 to - | -0.007 | -0.004
-0.012 to |
| Chapais      | 0.001            | 115        | 0.001            | -0.0003     | 0.0002         | 115        | 0.005           | 0.001       | 0.007  | -0.002              |
| Frelighsburg | 0.020            | ns         | 0.001            | ns          | 0.001          | ns         | 0.002           | ns          | -0.012 | ns                  |
| Sutton       | -0.029           | -0.058     | -0.002           | ns          | -0.001         | ns         | 0.002           | ns          | -0.006 | ns                  |
|              |                  | to -       |                  |             |                |            |                 |             |        |                     |
|              |                  | 0.005      |                  |             |                |            |                 |             |        |                     |
| Lac Edouard  | -0.004           | ns         | -0.002           | ns          | -0.002         | ns         | -0.014          | -0.022 to - | -0.025 | -0.046 to           |
|              |                  |            |                  |             |                |            |                 | 0.001       |        | -0.008              |
| Montmorency  | -0.023           | ns         | -0.002           | ns          | 0.001          | ns         | 0.009           | 0.0009 to   | 0.006  | ns                  |
|              |                  |            |                  |             |                |            |                 | 0.016       |        |                     |
| Kejimkujik   | 0.007            | ns         | 0.003            | ns          | -0.0003        | ns         | 0.024           | ns          | 0.056  | ns                  |

Table 3: Rate of change in annual wet deposition of  $Ca^{2+}$ ,  $Mg^{2+}$ ,  $K^+$ ,  $Na^+$ , and  $Cl^-$ . Slope refers to the Sen's slope (kg ha-1 a-1); C.I. refers to the 90% confidence interval of the slope; ns indicates no significant trend.

|              |                   |          |                              |                  |             |                  | nss-        |          | Annual precip |          |              |                   |
|--------------|-------------------|----------|------------------------------|------------------|-------------|------------------|-------------|----------|---------------|----------|--------------|-------------------|
| Site         | $\mathrm{NH_4}^+$ |          | NO 3 - |                  | $SO_4^{2-}$ |                  | $SO_4^{2-}$ |          | $(mm a^{-1})$ |          | $pH(a^{-1})$ |                   |
|              | Slope             | C.I.     | Slope                        | C.I.             | Slope       | C.I.             | Slope       | C.I.     | Slope         | C.I.     | Slope        | C.I.              |
| Saturna      | -0.01             | ns       | -0.07                        | -0.13 to         | -0.13       | -0.17 to         | -0.12       | -0.16 to | -1.5          | ns       | 0.012        | 0.009 to          |
|              |                   |          |                              | -0.01            |             | -0.06            |             | -0.07    |               |          |              | 0.016             |
| Esther       | 0.02              | ns       | 0.04                         | ns               | -0.02       | ns               | na          | na       | -1.4          | ns       | -0.012       | ns                |
| Cree Lake    | -0.02             | ns       | -0.05                        | ns               | -0.09       | ns               | na          | na       | -2.0          | ns       | -0.010       | ns                |
| Bratt's Lake | 0.07              | ns       | 0.02                         | ns               | 0.10        | ns               | na          | na       | 25.9          | ns       | -0.009       | ns                |
| FIΔ          | 0.05              | 0.03 to  | 0.02                         | ne               | -0.04       | ne               | na          | na       | 6.0           | ne       | 0.009        | 0.003 to          |
| LLA          | 0.05              | 0.05 10  | 0.02                         | 115              | -0.04       | 115              | na          | na       | 0.0           | 115      | 0.007        | 0.005 10          |
| Algoma       | -0.06             | -0.09 to | -0.38                        | -0.51 to         | -0.60       | -0.71 to         | na          | na       | -7.8          | -14.5 to | 0.020        | 0.014
0.016 to |
| 8            |                   | -0.02    |                              | -0.26            |             | -0.47            |             |          |               | -2.7     |              | 0.025             |
| Longwoods    | 0.03              | ns       | -0.33                        | -0.48 to         | -0.55       | -0.7 to          | na          | na       | 3.9           | ns       | 0.023        | 0.019 to          |
| -            |                   |          |                              | -0.17            |             | -0.36            |             |          |               |          |              | 0.028             |
| Egbert       | -0.001            | ns       | -0.31                        | -0.45 to         | -0.45       | -0.57 to         | na          | na       | 3.8           | ns       | 0.027        | 0.022 to          |
| _            |                   |          |                              | -0.18            |             | -0.33            |             |          |               |          |              | 0.03              |
| Sprucedale   | -0.03             | ns       | -1.01                        | -1.4 to          | -1.00       | -1.46 to         | na          | na       | 9.1           | ns       | 0.030        | 0.018 to          |
|              | 0.01              |          | 0.10                         | -0.48            | 0.26        | -0.36            |             |          | 5.0           | 0.1.4    | 0.000        | 0.042             |
| Chalk River  | 0.01              | ns       | -0.19                        | -0.24 to         | -0.36       | -0.43 to         | na          | na       | 5.3           | 2.1 to   | 0.020        | 0.018 to          |
| Chanaia      | 0.01              |          | 0.21                         | -0.11            | 0.24        | -0.29            |             |          | 0.0           | 8.3      | 0.015        | 0.022             |
| Chapais      | -0.01             | ns       | -0.21                        | -0.28 to         | -0.54       | -0.45 to         | na          | na       | -0.9          | ns       | 0.015        | 0.013 to          |
| Frelighsburg | 0.02              | ne       | 0.03                         | -0.11
1.20 to | 0.03        | -0.25
1.57 to | no          | na       | 13.5          | ne       | 0.056        | 0.018
0.036 to |
| Trengilsburg | -0.02             | 115      | -0.95                        | -0.64            | -0.95       | -0.16            | na          | na       | 15.5          | 115      | 0.050        | 0.05010           |
| Sutton       | 0.05              | 0.01 to  | 0.01                         | -0.04
ns      | -0.57       | -0.9 to          | na          | na       | 4.0           | ns       | 0.017        | 0.009             |
| Sutton       | 0100              | 0.09     | 0.01                         | 110              | 0.007       | -0.23            |             |          |               |          | 01017        | 0.026             |
| Lac Edouard  | -0.07             | ns       | -0.69                        | -0.91 to         | -0.46       | -0.77 to         | na          | na       | 14.6          | ns       | 0.027        | 0.018 to          |
|              |                   |          |                              | -0.38            |             | -0.23            |             |          |               |          |              | 0.045             |
| Montmorency  | 0.05              | ns       | 0.20                         | ns               | -0.27       | ns               | -0.27       | ns       | 27.8          | 6.1 to   | 0.008        | 0.001 to          |
| ·            |                   |          |                              |                  |             |                  |             |          |               | 45.7     |              | 0.014             |
| Kejimkujik   | 0.01              | ns       | -0.12                        | -0.18 to         | -0.27       | -0.37 to         | -0.28       | -0.36 to | 10.9          | 3 to     | 0.014        | 0.011 to          |
|              |                   |          |                              | -0.06            |             | -0.2             |             | -0.2     |               | 17.2     |              | 0.017             |

Table 4: Rate of change in annual wet deposition of  $NH_4^+$ ,  $NO_3^-$ ,  $SO_4^{-2-}$ , nss- $SO_4^{-2-}$ , precipitation amount and pH. Slope refers to the Sen's slope (kg ha-1 a-1); C.I. refers to the 90% confidence interval of the slope; ns indicates no significant trend; na indicates no available data.

---

## Referee Report (RR1)

I have read the revised version of the manuscript in detail and with great interest. I greatly appreciate the authors making the effort to address reviewer comments. I believe that the manuscript is now much strengthened and readable. Below I have listed some comments, which in my opinion still require some attention.

**Specific comments**

Both in the abstract and the text it is stated that the study analyzed long-term air concentrations, wet deposition, and precipitation acidity at 30 Canadian sites. The list of sites in Table 1, and Figure 6 and 8, however, include a total of up to 31 sites. Figure S1 shows the location of 29 sites only. These discrepancies are probably because of the duplicate stations in Egbert and Goose Bay, so, is 31 the final number of sites? If so please correct that number in the abstract, in the caption of Figure S1, and throughout the text.

Furthermore, differences in the availability of measurements among sites and years limited the comparison of results to a shorter number of locations (those with more than 9 years of data). As a result, the data presented in figures and tables, as well as the calculated averages, ranges, and trends mentioned and compared throughout the text, do not include results from all 31 locations in a consistent manner. For example, 16 sites were used for rates of change in annual air concentration/wet deposition data shown in tables 1, 2, and 3, while 12-14 sites were used for geographical patterns in air concentrations in figure 1, or 31 sites for wet deposition in figure 6. I imagine that for those same problems with data availability, the X axes of left and right panels in Figure 1 do not correspond to the same stations. Cree Lake and Montmorency data is shown only in 1983-1996 panels, while Bratt's Lake, Sprucedale, Frelighsburg, and Lac Edouard data appear in 1997-2010 panels only. Please summarize in the methods section the number of stations and/or time period used for each of the analyses on each of the subsections, as well as clarify the number of sites used for the calculation of averages and ranges presented in the text.

I am still not comfortable with the correlation analyses carried out between meteorological variables and particulate ions and trace gas data presented in page 10 (lines 14 to 34). I do not think that monthly averaged data correlations (or lack of) provide much information about the influences of temperature, precipitation, or relative humidity on observed K+ and NH4+ long-term trends. The use of Pearson correlation coefficients based on monthly averages to explain the long-term trends based on annual values seems a priori not very convincing. This is a complicated issue, as we are dealing with substances that differ in nature, origin, chemistry, and interactions with other pollutants. There is much evidence that current and future climatic variability and trends modulate the magnitude of annual emissions for substances like ammonia (Sutton et al. 2013). It is possible that Pearson correlation analysis of monthly data was not the best choice here. Considering that these analyses do not contribute much to the overall manuscript focus and goals, nor essential to support authors' main conclusions, I would suggest to eliminate these paragraphs in the final version of the manuscript. If necessary to support some of the statements authors made, reader can always be referred to the supplementary data. Doing so authors might improve clarity and it will also help reduce the sometimes "overwhelming" amount of data presented in the text, a concern that I expressed in my first round of comments.

**Additional comments**

Page 5, line 22: What do authors mean by "insufficient data"? Please clarify.

Page 7, line 10; Page 8, line 4; and elsewhere: authors refer to Fig. 1a and 1b while there are no such a and b panels, or they have not been labeled. Maybe left and right?

Figure 1: a 3-page figure seems a bit excessive. First, many of the elements should be deleted. Y-axis title (air concentration ug m-3) is the same for all panels so it can be placed just once, centered on the left. Y-axis tick labels and scale are identical between left and right panels so they do not need to be present on the right panel axis. X-axis labels are the same for the different substances so they only need to be shown on the bottom panel X-axes. Ion/trace gas labels only need to be shown in one of the two panels and not both. These adjustments will reduce the size of figure 1. Second, as a suggestion, authors might want to consider combining both panels in one single graph per substance, showing boxes of different colors (one color for 1983-1996 and another color for 1997-2010, stations that have data for one period only will show one box only).

Figure 6: As said for figure 1, please allocate X-axis labels on the bottom panel only, and just one Y-axis title centered.

**References cited**

Sutton, M. A., S. Reis, S. N. Riddick, U. Dragosits, E. Nemitz, M. R. Theobald, Y. S. Tang, C. F. Braban, M. Vieno, A. J. Dore, R. F. Mitchell, S. Wanless, F. Daunt, D. Fowler, T. D. Blackall, C. Milford, C. R. Flechard, B. Loubet, R. Massad, P. Cellier, E. Personne, P. F. Coheur, L. Clarisse, M. Van Damme, Y. Ngadi, C. Clerbaux, C. A. Skjøth, C. Geels, O. Hertel, R. J. Wichink Kruit, R. W. Pinder, J. O. Bash, J. T. Walker, D. Simpson, L. Horváth, T. H. Misselbrook, A. Bleeker, F. Dentener, and W. de Vries. 2013. Towards a climate-dependent paradigm of ammonia emission and deposition. Philosophical transactions of the Royal Society of London. Series B, Biological sciences 368:20130166.

---

## Author Response (AR2)

Dear Prof. Yu (Editor):

We are submitting a second revision of our paper entitled, "Long-term air concentrations, wet deposition, and scavenging ratios of inorganic ions, $HNO_3$ and $SO_2$ and assessment of aerosol and precipitation acidity at Canadian rural locations", for potential publication in Atmospheric Chemistry and Physics.  We have addressed all the additional comments provided by reviewer 1 on our revised manuscript.  Please see enclosed responses and track changes version of the paper.

Thank you for taking care of the review process.

Sincerely,

Irene Cheng and Leiming Zhang

Air Quality Research Division
Environment and Climate Change Canada

**Response to Reviewer 1 – Second Review**

I have read the revised version of the manuscript in detail and with great interest. I greatly appreciate the authors making the effort to address reviewer comments. I believe that the manuscript is now much strengthened and readable. Below I have listed some comments, which in my opinion still require some attention.

We appreciate the additional detailed comments that this reviewer provided to further improve and condense the materials in our paper. Our responses to the comments are provided below.

Specific comments
Both in the abstract and the text it is stated that the study analyzed long-term air concentrations, wet deposition, and precipitation acidity at 30 Canadian sites. The list of sites in Table 1, and Figure 6 and 8, however, include a total of up to 31 sites. Figure S1 shows the location of 29 sites only. These discrepancies are probably because of the duplicate stations in Egbert and Goose Bay, so, is 31 the final number of sites? If so please correct that number in the abstract, in the caption of Figure S1, and throughout the text.

Response: There are 31 sites; however there are two sets of precipitation concentrations at Egbert that are co-located. Thus, we stated there were actually 30 sites. Note that the two Goose Bay measurements are not exactly co-located as indicated by the slight differences in the geographical coordinates in Table 1. To ensure the number of sites is consistent, we changed the number of sites to 31 (now including the co-located measurement at Egbert) and added Egbert-2 and Goose Bay B to the map in Fig. S1.

Furthermore, differences in the availability of measurements among sites and years limited the comparison of results to a shorter number of locations (those with more than 9 years of data). As a result, the data presented in figures and tables, as well as the calculated averages, ranges, and trends mentioned and compared throughout the text, do not include results from all 31 locations in a consistent manner. For example, 16 sites were used for rates of change in annual air concentration/wet deposition data shown in tables 1, 2, and 3, while 12-14 sites were used for geographical patterns in air concentrations in figure 1, or 31 sites for wet deposition in figure 6. I imagine that for those same problems with data availability, the X axes of left and right panels in Figure 1 do not correspond to the same stations. Cree Lake and Montmorency data is shown only in 1983-1996 panels, while Bratt's Lake, Sprucedale, Frelighsburg, and Lac Edouard data appear in 1997-2010 panels only. Please summarize in the methods section the number of stations and/or time period used for each of the analyses on each of the subsections, as well as clarify the number of sites used for the calculation of averages and ranges presented in the text.

Response: This information has been provided in the paper already. In sect. 2.1.2 of the Methods section, we mentioned that air concentrations were available at 16 sites. In sect. 2.1.3, we

mentioned that precipitation concentrations were available at 30 sites (two co-located collectors at Egbert). Note the change to 31 sites in the revised paper. Also in section 2.1.1, we stated that the measurement periods are not synchronized between all sites with some sites having different start and end dates. The data coverages for each site were provided in Table 1.

I am still not comfortable with the correlation analyses carried out between meteorological variables and particulate ions and trace gas data presented in page 10 (lines 14 to 34). I do not think that monthly averaged data correlations (or lack of) provide much information about the influences of temperature, precipitation, or relative humidity on observed K+ and NH4+ long-term trends. The use of Pearson correlation coefficients based on monthly averages to explain the long-term trends based on annual values seems a priori not very convincing. This is a complicated issue, as we are dealing with substances that differ in nature, origin, chemistry, and interactions with other pollutants. There is much evidence that current and future climatic variability and trends modulate the magnitude of annual emissions for substances like ammonia (Sutton et al. 2013). It is possible that Pearson correlation analysis of monthly data was not the best choice here. Considering that these analyses do not contribute much to the overall manuscript focus and goals, nor essential to support authors' main conclusions, I would suggest to eliminate these paragraphs in the final version of the manuscript. If necessary to support some of the statements authors made, reader can always be referred to the supplementary data. Doing so authors might improve clarity and it will also help reduce the sometimes "overwhelming" amount of data presented in the text, a concern that I expressed in my first round of comments.

Response: This paragraph has been moved to sect. S1 of the Supplementary Material which supports the time-series graphs (Fig. S2) and the correlation analysis results (Table S1) that were already in the supplement. The last sentences of sect. S1 have been revised based on the reviewer's comments and the references by Sutton et al. (2013) and Yao and Zhang (2016). Sutton et al. (2013) suggests an increase in temperature by a few degrees would increase ammonia emissions. Yao and Zhang (2016) also suggest increasing ammonia emission with increasing temperature and provided two mechanisms: (1) ammonia emissions from soil would increase with increasing temperature, and (2) an increase in temperature would increase ammonium nitrate partitioning to gas-phase ammonia. Yao and Zhang (2016) further pointed out that the simultaneous decrease in sulfur dioxide emissions would likely reduce atmospheric sulfate and subsequently lead to lower ammonium sulfate. Thus, there are uncertainties on the effects of meteorological variables like temperature on the long-term sulfate and ammonium trends. The last sentences now reads, "The correlation analysis using monthly data did not find a strong relationship between long-term temperature changes and long-term trends in $SO_4^{2-}$ and $NH_4^+$ concentrations. The lack of trends is related to the combined effects of increasing temperature and decreasing sulfur dioxide emissions. Studies suggest an increase in temperature would increase ammonia emissions and the partitioning of ammonium nitrate to ammonia (Sutton et al., 2013; Yao and Zhang, 2016). However, the decreasing trend in sulfur dioxide

emissions would likely reduce atmospheric sulfate and subsequently lead to lower ammonium sulfate production (Yao and Zhang, 2016)."

Additional comments
Page 5, line 22: What do authors mean by "insufficient data"? Please clarify.

Response: To clarify, we revised the sentence to, "If there is insufficient data (<15 daily measurements) in each month, the scavenging ratio is not calculated."

Page 7, line 10; Page 8, line 4; and elsewhere: authors refer to Fig. 1a and 1b while there are no such a and b panels, or they have not been labeled. Maybe left and right?

Response: The labeling of the part a and b figure is incorrect.  Figure 1 has been revised.  There are no longer part a and b panels in the revised figure.

Figure 1: a 3-page figure seems a bit excessive. First, many of the elements should be deleted. Y-axis title (air concentration ug m-3) is the same for all panels so it can be placed just once, centered on the left. Y-axis tick labels and scale are identical between left and right panels so they do not need to be present on the right panel axis. X-axis labels are the same for the different substances so they only need to be shown on the bottom panel X-axes. Ion/trace gas labels only need to be shown in one of the two panels and not both. These adjustments will reduce the size of figure 1. Second, as a suggestion, authors might want to consider combining both panels in one single graph per substance, showing boxes of different colors (one color for 1983-1996 and another color for 1997-2010, stations that have data for one period only will show one box only).

Response: We revised the graphs according to both your suggestions.  The figure has been condensed to 1 page.  The white box graph is for the 1983-1996 data and the green box graph is for the 1997-2010 data (site labels also shown in green for this period).

Figure 6: As said for figure 1, please allocate X-axis labels on the bottom panel only, and just one Y-axis title centered.

Response: Revised according to your suggestions.  The figure has been condensed to 1 page.

References cited
Sutton, M. A., S. Reis, S. N. Riddick, U. Dragosits, E. Nemitz, M. R. Theobald, Y. S. Tang, C. F. Braban, M. Vieno, A. J. Dore, R. F. Mitchell, S. Wanless, F. Daunt, D. Fowler, T. D. Blackall, C. Milford, C. R. Flechard, B. Loubet, R. Massad, P. Cellier, E. Personne, P. F. Coheur, L. Clarisse, M. Van Damme, Y. Ngadi, C. Clerbaux, C. A. Skjøth, C. Geels, O. Hertel, R. J.

Wichink Kruit, R. W. Pinder, J. O. Bash, J. T. Walker, D. Simpson, L. Horváth, T. H. Misselbrook, A. Bleeker, F. Dentener, and W. de Vries. 2013. Towards a climate-dependent paradigm of ammonia emission and deposition. Philosophical transactions of the Royal Society of London. Series B, Biological sciences 368:20130166.

Yao, X. and Zhang, L.: Trends in atmospheric ammonia at urban, rural, and remote sites across North America, Atmos. Chem. Phys., 16, 11465-11475, doi:10.5194/acp-16-11465-2016, 2016.

[revised manuscript text omitted]

Fig. 2

[Figure]

[Figure]

Fig. 3

[Figure]

Fig. 4

[Figure]

Fig. 5

[Figure]

[Figure]

Fig. 6

[Figure]

Fig. 7

[Figure]

Fig. 8

[Figure]

Fig. 9

[Figure]

Fig. 10

[Figure]

*Supplementary Material*

**Long-term air concentrations, wet deposition, and scavenging ratios of inorganic ions, HNO$_3$ and SO$_2$ and assessment of aerosol and precipitation acidity at Canadian rural locations**

Irene Cheng and Leiming Zhang

Air Quality Research Division, Science and Technology Branch, Environment and Climate Change Canada, 4905 Dufferin Street, Toronto, Ontario, M3H 5T4, Canada

*Correspondence to*: Irene Cheng (irene.cheng@canada.ca)

**Section S1: Correlation analysis between particulate ions and meteorological variables**

The influence of meteorological parameters including temperature, relative humidity and precipitation rates on the temporal trends of particulate ions and trace gases were also investigated by performing correlation analyses on the monthly averaged data. The descending trend in $K^+$ between 1993 and 2010 at the majority of the sites was not strongly influenced by precipitation as evident by the weak correlation coefficients (Table S1). Higher correlation between monthly average $K^+$ and temperature were found at Sprucedale, Frelighsburg and Lac Edouard (r = 0.52-0.69, p<0.05). At these sites, the monthly average $K^+$ peaked in March-April and was at the minimum concentration during December-January, which resembled the seasonal temperature cycle (Fig. S2a). The higher $K^+$ in the early spring could be attributed to increase soil emissions from agriculture operations and forest fires during springtime since the major sources of particulate $K^+$ are from biomass and soil. Decreasing $NH_4^+$ observed at most of the sites was only weakly correlated with monthly precipitation rates and relative humidity, implying these meteorological parameters had little influence on the long-term temporal trend. Higher correlation between monthly average $NH_4^+$ and temperature was found at Kejimkujik (r = 0.63, p<0.05). The maximum $NH_4^+$ typically occurred during April-May and reached its lowest concentration during December-January (Fig. S2b). This seasonal trend is linked to the formation of $SO_4^{2-}$ through $SO_2$ oxidation, which tends to occur at higher temperatures because of increase production of atmospheric oxidants. This theory is consistent with the very high correlation between monthly average $NH_4^+$ and $SO_4^{2-}$ (r = 0.91, p<0.05) at Kejimkujik. Overall, strong correlations between $NH_4^+$ and $SO_4^{2-}$ were observed at all the sites (r = 0.6-0.94, p<0.05). The high correlation between $SO_4^{2-}/SO_2$ ratio and temperature (r = 0.61-0.84, p<0.05) suggests $SO_4^{2-}$ formation from the gas-phase oxidation of $SO_2$ (Yao et al., 2002). Besides the relationship between $NH_4^+$ and $SO_4^{2-}$, three of the sites including Saturna, ELA and Egbert exhibited strong correlations between $NH_4^+$ and $NO_3^-$ (r = 0.52-0.7, p<0.05) indicating that $NH_4^+$ also followed the temporal trend of $NO_3^-$ at some locations. In summary, precipitation and relative humidity had little impact on the long-term temporal patterns of particulate ions. Seasonal temperature trends  are linked with the seasonal cycle of atmospheric oxidants which explains the short-term patterns in $SO_4^{2-}$ and $NH_4^+$.  However, the correlation analysis using monthly data did not find a strong relationship between long-term temperature changes and long-term trends in $SO_4^{2-}$ and $NH_4^+$ concentrations. The lack of trends is related to the combined effects of increasing temperature and decreasing sulfur dioxide emissions. Studies suggest an increase in temperature would increase ammonia emissions and the partitioning of ammonium nitrate to ammonia (Sutton et al., 2013; Yao and Zhang, 2016). However, the decreasing trend in sulfur dioxide emissions would likely reduce atmospheric sulfate and subsequently lead to lower ammonium sulfate production (Yao and Zhang, 2016).

[Figure]

**Figure S1: Map of  31 CAPMoN sites in western/central Canada (a), southern Ontario (b), and Quebec and eastern Canada (c).**

[Figure]

**Figure S2: Time-series plots of monthly average air temperature and atmospheric K$^+$ from Sprucedale, Frelighsburg and Lac Edouard (combined) (a) and monthly average air temperature and atmospheric NH$_4^+$ at Kejimkujik (b).**

[Figure]

**Figure S3: Geographical pattern of the annual precipitation amount.  The blue line indicates the median; the red dot indicates the mean; the box and whiskers include the interquartile range and the 5th to 95th percentile range, respectively; the dotted line is the overall median among the sites.**

[Figure]

**Figure S4: Variability in scavenging ratios among different inorganic ions and trace gases within the same month for five different years at five of the sites.**

[Figure]

**Figure S5: Monthly variation in scavenging ratios among 13 sites. Error bars represent the standard deviation of scavenging ratios between sites. W$_{fPM}$ and W$_{cPM}$ are the fine and coarse particle scavenging ratios, respectively.**

[Figure]

**Figure S6. Monthly variation in SO₂ air concentrations. Error bars represent the standard deviation between sites.**

Table S1: Pearson correlation coefficients between selected ions and meteorological parameters (significant at p<0.05; otherwise non-significant (ns)).  Note that the ion concentrations and meteorological parameters are monthly averages.

| Site | $K^+$ | | | | $NH_4^+$ | | | | | | $SO_4^{2-}/SO_2$ |
|---|---|---|---|---|---|---|---|---|---|---|---|
| | Temp | RH | Precip | | Temp | RH | Precip | $NO_3^-$ | $SO_4^{2-}$ | | Temp |
| Saturna | -0.20 | 0.18 | 0.18 | | -0.04$^{ns}$ | -0.23 | -0.17 | 0.67 | 0.60 | | 0.61 |
| Bratt's Lake | 0.09$^{ns}$ | -0.20 | -0.25 | | | | | | | | |
| ELA | 0.20 | -0.03$^{ns}$ | 0.12 | | -0.26 | 0.13 | -0.22 | 0.70 | 0.78 | | 0.75 |
| Algoma | 0.42 | -0.07$^{ns}$ | -0.04$^{ns}$ | | 0.36 | 0.10 | -0.17 | 0.23 | 0.87 | | 0.73 |
| Longwoods | | | | | -0.05$^{ns}$ | 0.20 | -0.06$^{ns}$ | 0.44 | 0.68 | | 0.81 |
| Egbert | | | | | -0.07$^{ns}$ | 0.03$^{ns}$ | -0.21 | 0.52 | 0.75 | | 0.77 |
| Sprucedale | 0.69 | 0.05$^{ns}$ | -0.14$^{ns}$ | | 0.30 | 0.06$^{ns}$ | -0.23 | 0.20$^{ns}$ | 0.92 | | 0.73 |
| Chalk River | 0.31 | -0.005$^{ns}$ | 0.05$^{ns}$ | | 0.37 | 0.09 | 0.02$^{ns}$ | 0.10$^{ns}$ | 0.85 | | 0.64 |
| Chapais | 0.41 | -0.17 | 0.18 | | 0.38 | -0.21 | 0.04$^{ns}$ | 0.23 | 0.85 | | 0.77 |
| Frelighsburg | 0.53 | 0.08$^{ns}$ | 0.13$^{ns}$ | | 0.11$^{ns}$ | 0.13$^{ns}$ | -0.13$^{ns}$ | 0.36 | 0.86 | | 0.84 |
| Lac Edouard | 0.52 | -0.15$^{ns}$ | 0.21 | | 0.34 | 0.01$^{ns}$ | 0.11$^{ns}$ | 0.23 | 0.94 | | 0.76 |
| Kejimkujik | 0.31 | -0.17 | -0.15 | | 0.63 | -0.30 | -0.32 | -0.16 | 0.91 | | 0.83 |

Table S2: Average cation and anion concentrations (neq m$^{-3}$) in particulate matter by site and year

| Site | Cation | Anion | | Year | Cation | Anion |
|---|---|---|---|---|---|---|
| Saturna | 0.047 | 0.045 | | 1994 | 0.074 | 0.066 |
| Esther | 0.052 | 0.033 | | 1995 | 0.071 | 0.063 |
| Bratt's Lake | 0.060 | 0.031 | | 1996 | 0.073 | 0.063 |
| ELA | 0.024 | 0.024 | | 1997 | 0.073 | 0.063 |
| Algoma | 0.024 | 0.028 | | 1998 | 0.076 | 0.064 |
| Longwoods | 0.133 | 0.108 | | 1999 | 0.075 | 0.066 |
| Egbert | 0.096 | 0.072 | | 2000 | 0.067 | 0.062 |
| Sprucedale | 0.024 | 0.024 | | 2001 | 0.078 | 0.068 |
| Chalk River | 0.028 | 0.034 | | 2002 | 0.073 | 0.061 |
| Chapais | 0.016 | 0.019 | | 2003 | 0.065 | 0.055 |
| Frelighsburg | 0.060 | 0.043 | | 2004 | 0.063 | 0.054 |
| Sutton | 0.062 | 0.054 | | 2005 | 0.069 | 0.059 |
| Lac Edouard | 0.016 | 0.018 | | 2006 | 0.058 | 0.046 |
| Montmorency | 0.023 | 0.026 | | 2007 | 0.058 | 0.048 |
| Kejimkujik | 0.035 | 0.041 | | 2008 | 0.060 | 0.048 |
| | | | | 2009 | 0.052 | 0.041 |
| | | | | 2010 | 0.048 | 0.037 |

Table S3: Scavenging ratio (W) statistics from this study (mass basis).  na indicates no available data. "C" indicates coastal locations.

| Site | $W_{Ca2+}$ | $W_{Mg2+}$ | $W_{Na+}$ | $W_{coarsePM}$ | $W_{finePM} = W_k$ | $W_{Cl}$ | $W_{NH4}$ | $W_{pNO3}$ | $W_{HNO3}$ | $W_{pSO4}$ | $W_{SO2}$ |
|---|---|---|---|---|---|---|---|---|---|---|---|
| **Mean** | | | | | | | | | | | |
| Saturna[C] | 1671 | 1493 | 1390 | 1505 | 567 | 2103 | 445 | 897 | 776 | 602 | 233 |
| ELA | 1893 | 1347 | 1208 | 1496 | 885 | 3519 | 848 | 1224 | 3079 | 696 | 1247 |
| Algoma | 2413 | 1510 | 1002 | 1426 | 911 | 5779 | 762 | 1204 | 2064 | 734 | 330 |
| Longwoods | 911 | 901 | 865 | 881 | 996 | 2011 | 303 | 657 | 749 | 601 | 125 |
| Egbert | 625 | 748 | 713 | 697 | 554 | 1914 | 283 | 583 | 982 | 473 | 181 |
| Sprucedale | 1822 | 1299 | 905 | 1342 | 611 | 4278 | 577 | 998 | 2012 | 611 | 445 |
| Chalk River | na | na | 666 | 666 | 580 | 5065 | 466 | 632 | 3005 | 547 | 560 |
| Chapais | 2077 | 899 | 475 | 1043 | 758 | 1855 | 625 | 932 | 5527 | 661 | 1291 |
| Frelighsburg | 852 | 838 | 647 | 781 | 553 | 2554 | 411 | 642 | 1375 | 471 | 414 |
| Sutton | 1010 | 936 | 594 | 753 | 497 | 4377 | 350 | 631 | 1616 | 473 | 396 |
| Lac Edouard | 1702 | 1105 | 577 | 1128 | 512 | 3192 | 465 | 842 | 2398 | 496 | 771 |
| Montmorency | 1200 | 1148 | 599 | 742 | 607 | 4969 | 406 | 697 | 2201 | 471 | 1222 |
| Kejimkujik[C] | 1729 | 1980 | 2120 | 2077 | 702 | 4959 | 303 | 1120 | 1842 | 571 | 1339 |
| Max avg/min avg ratio | 3.9 | 2.6 | 4.5 | 3.1 | 2.0 | 3.1 | 3.0 | 2.1 | 7.4 | 1.6 | 10.7 |
| **Median** | | | | | | | | | | | |
| Saturna[C] | 1552 | 1404 | 1245 | 1430 | 502 | 1934 | 416 | 796 | 650 | 533 | 208 |
| ELA | 1395 | 1141 | 993 | 1358 | 765 | 3235 | 726 | 1151 | 2711 | 628 | 1141 |
| Algoma | 1946 | 1266 | 816 | 1272 | 769 | 4532 | 655 | 1046 | 1880 | 698 | 203 |
| Longwoods | 840 | 832 | 775 | 802 | 682 | 1684 | 252 | 601 | 857 | 531 | 60 |
| Egbert | 535 | 693 | 585 | 635 | 440 | 1355 | 245 | 521 | 829 | 428 | 127 |
| Sprucedale | 1504 | 1238 | 765 | 1218 | 556 | 3872 | 582 | 945 | 1906 | 582 | 204 |

| | | | | | | | | | | | |
|---|---|---|---|---|---|---|---|---|---|---|---|
| Chalk River | na | na | 531 | 531 | 437 | 4321 | 401 | 526 | 2568 | 449 | 284 |
| Chapais | 1743 | 712 | 363 | 883 | 640 | 1216 | 544 | 787 | 4914 | 578 | 828 |
| Frelighsburg | 774 | 754 | 540 | 742 | 438 | 1788 | 405 | 599 | 1322 | 442 | 251 |
| Sutton | 968 | 888 | 549 | 706 | 445 | 2972 | 314 | 562 | 1452 | 435 | 241 |
| Lac Edouard | 1483 | 1015 | 497 | 1023 | 450 | 2480 | 380 | 780 | 2077 | 404 | 453 |
| Montmorency | 1130 | 833 | 474 | 623 | 432 | 3633 | 347 | 559 | 2173 | 386 | 809 |
| Kejimkujik[C] | 1482 | 1416 | 1447 | 1452 | 541 | 3511 | 279 | 926 | 1571 | 488 | 461 |
| Max median/min median ratio | 3.6 | 2.0 | 4.0 | 2.7 | 1.8 | 3.7 | 3.0 | 2.2 | 7.6 | 1.8 | 19.0 |
| **Standard deviation** | | | | | | | | | | | |
| Saturna[C] | 798 | 727 | 734 | 655 | 271 | 1001 | 217 | 381 | 547 | 276 | 123 |
| ELA | 1684 | 810 | 1016 | 990 | 483 | 2159 | 476 | 680 | 1636 | 336 | 880 |
| Algoma | 1869 | 1061 | 661 | 973 | 653 | 5161 | 517 | 753 | 1017 | 284 | 372 |
| Longwoods | 439 | 398 | 362 | 312 | 869 | 1331 | 159 | 233 | 453 | 273 | 142 |
| Egbert | 373 | 317 | 427 | 322 | 501 | 1547 | 170 | 307 | 647 | 214 | 180 |
| Sprucedale | 1039 | 634 | 637 | 597 | 286 | 2940 | 253 | 504 | 840 | 259 | 523 |
| Chalk River | na | na | 450 | 450 | 417 | 3763 | 313 | 392 | 1629 | 352 | 681 |
| Chapais | 1273 | 668 | 349 | 641 | 407 | 1969 | 328 | 529 | 2654 | 339 | 1373 |
| Frelighsburg | 415 | 367 | 382 | 331 | 514 | 2288 | 157 | 311 | 523 | 170 | 438 |
| Sutton | 374 | 371 | 254 | 302 | 321 | 3320 | 155 | 284 | 618 | 234 | 445 |
| Lac Edouard | 1003 | 597 | 361 | 517 | 281 | 2188 | 257 | 412 | 1195 | 260 | 843 |
| Montmorency | 628 | 783 | 354 | 441 | 478 | 4489 | 231 | 425 | 960 | 324 | 1151 |
| Kejimkujik[C] | 1134 | 1643 | 1942 | 1823 | 671 | 4723 | 154 | 687 | 1172 | 391 | 2245 |

Table S4: Scavenging ratios from literature (mass basis). * Most literature values excluded gas scavenging except where indicated; [1]Derived from multiple linear regression; [2]$Wp_{NO3}$ based on sum of $pNO_3^-$ and $HNO_3$ in air; [3]$W_{pSO4}$ based on sum of $pSO_4^{2-}$ and $SO_2$ in air; [4]Snow events only.

| Location | $W_{Ca2+}$ | $W_{Mg2+}$ | $W_{Na+}$ | $W_{K+}$ | $W_{Cl}$ | $*W_{NH4}$ | $*W_{pNO3}$ | $*W_{pSO4}$ | $W_{HNO3}$ | $W_{SO2}$ | Reference |
|---|---|---|---|---|---|---|---|---|---|---|---|
| CAPMoN sites, Canada | 625-2413 | 748-1980 | 475-2120 | 497-996 | 1855-5779 | 283-848 | 583-1224 | 471-734 | 749-5527 | 125-1339 | This study |
| | | | | | | | 523-1776 | 391-903 | 722-2848 | 31-409 | This study[1] |
| Eastern Canada | | | | | | | 832-2950 | 831-1550 | | | Barrie et al. (1985)[2] |
| NADP/CASTNET sites, U.S. | 860-2526 | 656-2011 | 798-7409 | 265-885 | 1430-22950 | | | | | | Cheng et al. (2015) |
| NADP/AIRMoN sites, U.S. | | | | | | 96-3050 | | 216-2710 | 300-1700 | | Hicks (2005)[3] |
| Northern Michigan | | | | 80-4600 | | 40-2200 | 500 | 60-2280 | 3500-4550 | 219-355 | Cadle et al. (1990)[4] |
| Barbados | | | 578-869 | | | | 336-409 | 264-315 | | | Galloway et al. (1993) |
| Bermuda | | | 560-749 | | | | 273-447 | 182-242 | | | Galloway et al. (1993) |
| Ireland | | | 1692-5800 | | | | 535-1258 | 321-734 | | | Galloway et al. (1993) |
| Vitoria, Spain | 3983 | | 3625 | 1151 | 3030 | 1739 | 2303 | 2830 | | | Encinas et al. (2004) |
| Central France | | | | 20000-250000 | | 500-8500 | 4500-16500 | 750-5250 | | | Bourcier et al. (2012) |
| Sardinia, Italy | 3459-4605 | 1013-1524 | 1708-1815 | 1630-1845 | 2085-2419 | | 754-1062 | 1728-2173 | | | Guerzoni et al. (1995) |
| Turkey | 1649±466 | | 789±176 | 619±138 | | 624±147 | 612±228 | 1177±301 | | | Tuncel and Ungör (1996) |
| Mt. Sonnblick, Austria | | | | | | 2160 | 3120 | 1680 | | | Kasper-Giebl et al. (1998)[2] |
| | | | | | | | 1104 | 1056 | 3480 | 96 | Kasper-Giebl et al. (1998)[1] |
| Hyderabad, India | 2265 | 877 | 517 | 723 | 109 | 389 | 317 | 160 | | | Kulshrestha et al. (2009) |
| Bay of Bengal | 582 | 5039 | 5282 | 58 | 16085 | 336 | 3591 | 426 | | | Kulshrestha et al. (2009) |
| Nepal | 1805±1873 | 2580±2277 | 1393±1593 | 2436±2092 | 2582±1870 | 1811±2351 | 2502±2007 | 487±326 | | | Shrestha et al. (2002) |
| Singapore | 697±376 | 318±201 | 500±180 | 744±590 | 2624±1129 | 1660±1284 | 2134±1671 | 2596±2076 | | | He and Balasubramanian (2008) |
| Maldives | 470-1200 | 500 | 1000-1100 | 140-220 | 1100-1200 | 530-600 | 930-2000 | 490-580 | | | Granat et al. (2010) |

Table S5: Gas and particle scavenging ratios for nitrate and sulfate derived from multiple linear regression [$C_{prec} = \text{constant} + W_{gas}C_{gas,air} + W_{part}C_{part,air}$]. $C_{prec}$ is the total precipitation concentration of nitrate or sulfate; $W_{gas}$ and $W_{part}$ are the gas and particle scavenging ratios, respectively; $C_{gas,air}$ and $C_{part,air}$ are the air concentrations of $HNO_3$ or $SO_2$ and $pNO_3^-$ or $pSO_4^{2-}$, respectively. $R^2$ is the coefficient of determination and r-part is the partial correlation. ns indicates not significant coefficients ($p>0.05$).

| Site | $W_{pNO3}$ | $W_{HNO3}$ | $R^2$ | r-part $pNO_3^-$ | r-part $HNO_3$ | | $W_{pSO4}$ | $W_{SO2}$ | $R^2$ | r-part $pSO_4^{2-}$ | r-part $SO_2$ |
|---|---|---|---|---|---|---|---|---|---|---|---|
| Saturna | 910 | 1175 | 0.20 | 0.22 | 0.33 | | 903 | 45 | 0.14 | 0.31 | 0.06 |
| Bratt's Lake | 694 | 1359 | 0.16 | 0.29 | 0.25 | | | | <0.1 | | |
| Algoma | 1121 | 1253 | 0.24 | 0.25 | 0.28 | | 434 | 85 | 0.17 | 0.33 | 0.08 |
| Longwoods | 544 | 776 | 0.14 | 0.28 | 0.21 | | 393 | ns | 0.12 | 0.33 | -0.01 |
| Egbert | 523 | 722 | 0.18 | 0.30 | 0.20 | | 391 | ns | 0.16 | 0.37 | 0.01 |
| Sprucedale | 882 | 1308 | 0.21 | 0.21 | 0.26 | | 505 | ns | 0.13 | 0.35 | -0.02 |
| Chalk River | 786 | 1510 | 0.17 | 0.15 | 0.28 | | 487 | ns | 0.14 | 0.36 | -0.01 |
| Chapais | 1776 | 2848 | 0.25 | 0.16 | 0.39 | | 623 | 409 | 0.20 | 0.41 | -0.04 |
| Sutton | 944 | 1241 | 0.15 | 0.19 | 0.25 | | 535 | 31 | 0.17 | 0.39 | 0.03 |
| Lac Edouard | 1532 | 1289 | 0.33 | 0.25 | 0.22 | | 482 | 80 | 0.19 | 0.37 | 0.06 |
| Frelighsburg | 958 | 1470 | 0.26 | 0.28 | 0.29 | | 526 | 104 | 0.13 | 0.32 | 0.08 |
| Montmorency | 1470 | 1493 | 0.34 | 0.16 | 0.45 | | 481 | ns | 0.22 | 0.46 | -0.03 |
| Kejimkujik | 1561 | 2382 | 0.19 | 0.22 | 0.33 | | | | <0.1 | | |
| *Overall* | 570 | 1253 | 0.21 | 0.22 | 0.29 | | 489 | 32 | 0.20 | 0.38 | 0.04 |

Table S6: Monthly gas and particle scavenging ratios for nitrate and sulfate derived from multiple linear regression (similar to Table S5)

| Month | $W_{pNO3}$ | $W_{HNO3}$ | $R^2$ | r-part $pNO_3^-$ | r-part $HNO_3$ | | $W_{pSO4}$ | $W_{SO2}$ | $R^2$ | r-part $pSO_4^{2-}$ | r-part $SO_2$ |
|---|---|---|---|---|---|---|---|---|---|---|---|
| Jan | 425 | 1871 | 0.23 | 0.16 | 0.37 | | 851 | ns | 0.25 | 0.40 | -0.06 |
| Feb | 548 | 1511 | 0.27 | 0.24 | 0.35 | | 467 | 184 | 0.20 | 0.34 | 0.00 |
| Mar | 583 | 1283 | 0.26 | 0.26 | 0.31 | | 633 | ns | 0.18 | 0.32 | 0.02 |
| Apr | 442 | 1518 | 0.21 | 0.18 | 0.30 | | 525 | 195 | 0.22 | 0.28 | 0.09 |
| May | 270 | 1332 | 0.18 | 0.10 | 0.31 | | 434 | 257 | 0.19 | 0.22 | 0.13 |
| Jun | 373 | 1093 | 0.22 | 0.11 | 0.31 | | 284 | 307 | 0.22 | 0.22 | 0.17 |
| Jul | 636 | 1005 | 0.19 | 0.13 | 0.29 | | 317 | 204 | 0.21 | 0.26 | 0.09 |
| Aug | 894 | 1065 | 0.24 | 0.17 | 0.30 | | 333 | 230 | 0.20 | 0.26 | 0.09 |
| Sep | 345 | 842 | 0.11 | 0.08 | 0.20 | | 362 | 77 | 0.17 | 0.29 | 0.05 |
| Oct | 749 | 862 | 0.17 | 0.24 | 0.19 | | 472 | 107 | 0.24 | 0.28 | 0.11 |
| Nov | 712 | 1543 | 0.20 | 0.28 | 0.24 | | 780 | 21 | 0.21 | 0.35 | 0.03 |
| Dec | 554 | 1741 | 0.24 | 0.25 | 0.33 | | 670 | ns | 0.19 | 0.32 | 0.00 |